# Online Budgeted Matching with General Bids

**Jianyi Yang**
University of Houston
Houston, TX, USA
jyang71@central.uh.edu

**Pengfei Li**
University of California, Riverside
Riverside, CA, USA
pli081@ucr.edu

**Adam Wierman**
California Institute of Technology
Pasadena, CA, USA
adamw@caltech.edu

**Shaolei Ren**
University of California, Riverside
Riverside, CA, USA
shaolei@ucr.edu

## Abstract

Online Budgeted Matching (OBM) is a classic problem with important applications in online advertising, online service matching, revenue management, and beyond. Traditional online algorithms typically assume a small bid setting, where the maximum bid-to-budget ratio ($\kappa$) is infinitesimally small. While recent algorithms have tried to address scenarios with non-small or general bids, they often rely on the Fractional Last Matching (FLM) assumption, which allows for accepting partial bids when the remaining budget is insufficient. This assumption, however, does not hold for many applications with indivisible bids. In this paper, we remove the FLM assumption and tackle the open problem of OBM with general bids. We first establish an upper bound of $1 - \kappa$ on the competitive ratio for any deterministic online algorithm. We then propose a novel meta algorithm, called `MetaAd`, which reduces to different algorithms with first known provable competitive ratios parameterized by the maximum bid-to-budget ratio $\kappa \in [0, 1]$. As a by-product, we extend `MetaAd` to the FLM setting and get provable competitive algorithms. Finally, we apply our competitive analysis to the design learning-augmented algorithms.

## 1 Introduction

Online Budgeted Matching (OBM) with general bids is a fundamental online optimization problem that generalizes to many important settings, such as online bipartite matching and Adwords with equal bids [23]. It has applications in various domains, including online advertising, online resource allocation, and revenue management among others [5, 16, 32]. OBM is defined on a bipartite graph with a set of offline nodes (bidders) and a set of online nodes (queries). The task is to select an available offline node to match with an online query in each round. When an offline node is matched to an online node, a bid value is subtracted from the budget of the offline node, and a reward equal to the consumed budget is obtained. If the remaining budget of an offline node is less than the bid value of an online query, the offline node cannot be matched to the online query. The goal is to maximize the total reward throughout the entire online matching process.

OBM is challenging due to the nature of online discrete decisions. Previous works have studied this problem under one of the following two additional assumptions on bids or matching rules:
• *Small bids.* The small-bid assumption is a *special* case of general bids corresponding to the maximum bid-budget ratio $\kappa \to 0$. That is, while the bid values can vary arbitrarily, the size of each individual bid is infinitely small compared to each offline node's budget, and there is always enough budget for matching. Under this assumption, the first online algorithm was provided by [24],

achieving an optimal competitive ratio of $1 - 1/e$ [23]. This competitive ratio has also been attained by subsequent algorithms based on primal-dual techniques [4, 7]. However, the small-bid assumption significantly limits these algorithms for broader applications in practice. Take the application of matching Virtual Machines (VMs) to physical servers as an example. An online VM request typically takes up a non-negligible fraction of the total computing units in a server.

• *Fractional last match (FLM)*. Under FLM, if an offline node has an insufficient budget for an online query, the offline node can still be matched to the query, obtaining a partial reward equal to the remaining budget. Given the limitations of small bids, some recent studies [15, 29, 30] have studied competitive algorithms for OBM with general bids by making the additional assumption of FLM. For example, under FLM, the greedy algorithm (`Greedy`) achieves a competitive ratio of $1/2$, while other studies [4, 15, 29, 30] aim to achieve a competitive ratio greater than $1/2$ under various settings and/or using randomized algorithms. Although FLM allows fractional matching of a query to an offline node with insufficient budgets, it essentially assumes that any bids are potentially divisible. This assumption may not hold in many real applications, e.g., allocating fractional physical resources to a VM can result in significant performance issues that render the allocation unacceptable, and charging a fractional advertising fee may not be allowed in online advertising.

Despite its practical relevance and theoretical importance, OBM with general bids has remained a challenging open problem in the absence of the small-bid and FLM assumptions. Specifically, an offline node may have insufficient budget and cannot be matched to a later query with a large value, potentially causing large sub-optimality in the worst case. This issue does not apply to small bids, as the small-bid setting implies that insufficient budgets will never occur. Additionally, this challenge is alleviated in the FLM setting, where fractional matching in cases of insufficient budgets can reduce sub-optimality. Indeed, removing the small-bid and FLM assumptions fundamentally changes and add significant challenges to the problem of OBM [30]. To further highlight the intrinsic difficulty of OBM with general bids, we formally prove in Proposition 4.1 an upper bound of the competitive ratio, i.e., $1 - \kappa$, achieved by any deterministic online algorithm, where $\kappa \in [0, 1]$ is the maximum bid-budget ratio.

**Contributions**: In this paper, we address OBM *without* the *small-bid* or *FLM* assumptions and design a meta algorithm called `MetaAd`, which adapts to different algorithms with provable competitive ratios. To our knowledge, `MetaAd` is the first provable competitive algorithm for general bids without the FLM assumption. Specifically, `MetaAd` generates a discounted score for each offline node by a general discounting function, which is then used to select the offline node. The discounting function evaluates the degree of budget insufficiency given a bid-budget ratio $\kappa \in [0, 1]$, addressing the challenge of infeasible matching due to insufficient budgets. Given different discounting functions, `MetaAd` yields concrete algorithms, and their competitive ratios are derived from Theorem 4.2, established through a novel proof technique. We show that with small bids (i.e., $\kappa \to 0$), `MetaAd` recovers the optimal competitive ratio of $1 - \frac{1}{e}$. Furthermore, we show that `MetaAd`, with discounting functions from the exponential and polynomial function classes, achieves a positive competitive ratio for $\kappa \in [0, 1)$. As an extension, we adapt the design of `MetaAd` to the FLM setting, resulting in a meta-algorithm with provable competitive ratios for $\kappa \in [0, 1]$ (Theorem 4.3). The framework of `MetaAd` potentially opens an interesting direction for exploring concrete discounting function designs that yield high competitive ratios for settings both with and without FLM. Finally, we apply our competitive analysis to the design of `LOBM`, a learning-augmented algorithm for OBM, which enhances average performance while still guaranteeing a competitive ratio (Theorem 5.1). We validate the empirical benefits of `MetaAd` and `LOBM` through numerical experiments on the applications of an online movie matching an VM placement on physical servers.

## 2 Related Work

OBM originates from the online bipartite matching problem defined by [19] 30 years ago. In 2007, [24] generalized the online b-matching problem to OBM (a.k.a. Adwords) [17]. Under the special case of small bids, [24] proposes an algorithm that achieves the competitive ratio of $1 - 1/e$, which is also the optimal competitive ratio under the small-bid setting [17]. In the same year, [4] provides the primal-dual algorithm and analysis for OBM under the small-bid assumption and achieves the competitive ratio of $1 - \frac{1}{e}$. Subsequently, [7] gives a randomized primal-dual analysis for online bipartite matching and generalizes it to OBM. In addition, OBM has also been studied under the stochastic settings [9, 8, 6, 14, 25].

It is known to be very challenging to go beyond the small-bid assumption and develop a non-trivial competitive ratio for OBM with general bids in an adversarial setting. Recently, [30] points out the inherent difficulty of OBM with general bids and explains the necessity of the assumption of FLM is needed in easing up the challenges. With the FLM assumption, a greedy algorithm can achieve a competitive ratio of $1/2$. Additionally, a deterministic algorithm proposed in [4] achieves a competitive ratio of $(1 - \kappa - \frac{1-\kappa}{(1+\kappa)^{1/\kappa}})$, increasing the competitive ratio when the maximum bid-budget ratio $\kappa$ is no larger than $0.17$. Some other works on OBM with FLM employ randomized algorithm designs. For example, [15] proposes a semi-random algorithm that achieves a competitive ratio of $0.5016$, which is known as the best competitive ratio achieved by randomized algorithms up to now. Besides, [30] extends the random algorithm of Ranking to OBM and achieves a competitive ratio of $1 - 1/e$ with a strong assumption of the "fake" budget. Moreover, [29] proposes a randomized algorithm without the knowledge of budget for the FLM setting with competitive ratio $\frac{1}{1+\kappa}(1 - \frac{1}{e})$ parameterized by the maximum bid-budget ratios $\kappa$, which relies on a strong assumption that the bids are decomposable (i.e. $w_{u,t} = w_u \cdot w_t$).

Despite the progress on the OBM settings with FLM, OBM without FLM has remained an open challenge except for under the small-bid assumption. The recent study [30] points out that OBM without FLM is difficult because when the offline node has less leftover budget than the bid value of an online arrival, the offline node is not allowed to be matched to the arrival, potentially causing a loss equivalent the leftover budget. [20] considers multi-tier budget constraints with a laminar structure and provides a competitive ratio without FLM, but the result does not apply to the settings without the laminar structure. Additionally, [18] proves an online randomized algorithm with the competitive ratio (in expectation) upper bound of $0.612$, an upper bound for deterministic algorithms is still lacking to formally evaluate the difficulty of OBM without FLM.

## 3 Problem Formulation

We consider OBM with general bids. Specifically, there is a bipartite graph described as $G(\mathcal{U}, \mathcal{V}, E)$, where the vertices $u \in \mathcal{U}$ (i.e. offline nodes or bidders) are fixed and the vertices $v \in \mathcal{V}$ (i.e. online nodes or queries) arrive sequentially. The edge corresponding to vertices $u \in \mathcal{U}$ and $v \in \mathcal{V}$ has a bid value $w_{u,v} \geq 0$ which is the amount the offline node $u$ would like to pay for the online node $v$ if matched. The sizes of the vertex sets are denoted as $|\mathcal{U}| = U$ and $|\mathcal{V}| = V$, respectively. We index each online node by its arriving order, i.e. online node $t$ arrives at the $t$-th round.

At the beginning, each offline node $u \in \mathcal{U}$ has an initial budget $b_{u,0} = B_u \geq 0$, which is the maximum amount the offline node can pay in total. At round $t$, a query $t \in \mathcal{V}$ arrives, the edges connected to this arrival $t$ are revealed, and the agent chooses to match the arrival to an *available* offline node from $\mathcal{U}_t = \{u \in \mathcal{U} \mid w_{u,t} > 0, b_{u,t-1} \geq w_{u,t}\}$ or skip this query without any matching. We denote the agent's action as $x_t \in \mathcal{U}_t \bigcup \{\text{null}\}$, where "null" represents skipping this arrival. If at round $t$, an offline node $x_t$ is matched to the query $t$, a reward $r_t = w_{x_t,t}$ is earned and a bid value $w_{x_t,t}$ is charged from the offline node $x_t$, i.e., $b_{x_t,t} = b_{x_t,t-1} - w_{x_t,t}$; for the other offline nodes $u \neq x_t$, the budget remains unchanged $b_{u,t} = b_{u,t-1}$. If $x_t = \text{null}$, no reward is earned, i.e. $r_t = 0$ and the budgets of all offline nodes remain the same as the last round, i.e. $b_{u,t} = b_{u,t-1} \ \forall u \in \mathcal{U}$. The cumulative consumed budget at the end of round $t$ is denoted as $c_{u,t} = B_u - b_{u,t}, \forall u \in \mathcal{U}$ and $t \in [V]$. The agent aims to maximize the total reward $P = \sum_{t=1}^{V} r_t$ over the entire $V$ rounds.

The *offline* version of OBM can be written as Linear Programming (LP) with its primal and dual problems given in (1), where $P$ is the primal objective and $D$ is the dual objective. While we only need to solve the primal problem for OBM, we present the dual problem with dual variables $\alpha_u, u \in \mathcal{U}$ and $\beta_t, t \in \mathcal{V}$ to facilitate the subsequent algorithm design and analysis.

$$
\begin{aligned}
&\max P := \sum_{t=1}^{V} \sum_{u \in \mathcal{U}} w_{u,t} x_{u,t} && \min \ D := \sum_{u \in \mathcal{U}} B_u \alpha_u + \sum_{t=1}^{V} \beta_t \\
&\text{s.t. } \forall u \in \mathcal{U}, \sum_{t=1}^{V} w_{u,t} x_{u,t} \leq B_u, && \text{s.t. } \forall u \in \mathcal{U}, t \in [V], w_{u,t}\alpha_u + \beta_t \geq w_{u,t}, \\
&\quad \forall t \in [V], \sum_{u \in \mathcal{U}} x_{u,t} \leq 1, && \quad \forall u \in \mathcal{U}, \alpha_u \geq 0, \\
&\quad \forall u \in \mathcal{U}, v \in [V], x_{u,t} \geq 0. && \quad \forall t \in [V], \beta_t \geq 0.
\end{aligned}
\tag{1}
$$

A common performance metric for online algorithms is the **competitive ratio** defined as

$$\eta = \min_{G \in \mathcal{G}} \{P(G)/P^*(G)\}, \tag{2}$$

where the minimization is taken over the set of all possible bipartite graphs $\mathcal{G}$ [1], $P(G) = \sum_{t=1}^{V} w_{x_t,t}$ is the total reward obtained by an (online) algorithm for a graph $G \in \mathcal{G}$, and $P^*(G) = \sum_{t=1}^{V} w_{x_t^*,t}$ is the corresponding offline optimal total reward with $x_t^*$ being the offline optimal solution to (1).

Next, we formally define the bid-budget ratio $\kappa \in [0,1]$ in Definition 1 which is the maximum ratio of the bid value of an offline node to its total budget. We use $\eta(\kappa)$ to denote the competitive ratio of an algorithm for OBM with bid-budget ratio $\kappa$.

**Definition 1** (Bid-budget ratio). *The bid-budget ratio $\kappa \in [0,1]$ for an example $G(\mathcal{U}, \mathcal{V}, E)$ is defined as $\kappa = \sup_{u \in \mathcal{U}, t \in \mathcal{V}} \frac{w_{u,t}}{B_u}$.*

Many previous works [23, 24, 15, 4] assume FLM which allows for accepting partial bids when remaining budget is insufficient (i.e. modifying each bid $w_{u,t}$ to $\bar{w}_{u,t} = \min\{w_{u,t}, b_{u,t-1}\}$ given the remaining budget $b_{u,t-1}$). Without the FLM assumption, the only known competitive ratio for OBM is for the small-bid setting where $\kappa$ is infinitely small and approaches zero [23, 24]. However, the small-bid and FLM assumptions do not hold in many real-world applications as illustrated by the following examples:

• **Online VM placement**. In this problem, a cloud manager allocates virtual machines (VMs, online nodes) to heterogeneous physical servers (offline nodes), each with a computing resource capacity of $B_u'$ [10, 28]. When a VM request with a computing load of $z_t$ arrives, the manager assigns it to a server. If the VM is placed on server $u$, the manager receives a utility of $w_{u,v} = r_u z_v$ due to the heterogeneity of servers. The goal is to maximize the total utility $\sum_{t=1}^{V} \sum_{u \in \mathcal{U}} w_{u,t} x_{u,t}$ subject to the computing resource constraint $\sum_{t=1}^{V} z_t x_{u,t} \le B_u'$ for each server $u$, which can also be written as $\sum_{t=1}^{V} w_{u,t} x_{u,t} \le B_u$ with $B_u = r_u B_u'$. In this problem, VMs are not divisible and consume up a non-negligible portion of the server capacity, violating both the small-bid and FLM assumptions.

• **Inventory management with indivisible goods**. Here, a manager must match several indivisible goods (online nodes) to various resource nodes (offline nodes), each with a limited capacity (e.g., matching parcels to mail trucks or food orders to delivery vehicles). Each good can only be assigned to one node without being split, and a good $t$ can occupy a substantial portion of the resource node's capacity, $w_{u,t}$. The goal is to maximize the total utilization $\sum_{t=1}^{V} \sum_{u \in \mathcal{U}} w_{u,t} x_{u,t}$, subject to the capacity constraint $\sum_{t=1}^{V} w_{u,t} x_{u,t} \le 1$ for each node $u$. In this problem, neither the small-bid nor FLM assumption applies.

In this paper, *without* relying on the small-bid or FLM assumptions, we move beyond small bids and propose a meta algorithm (`MetaAd`) for general $\kappa \in [0,1]$ which can reduce to many concrete competitive algorithms.

## 4   `MetaAd`: Meta Algorithm for OBM

### 4.1   An Upper Bound on the Competitive Ratio

In the absence of small-bid and FLM assumptions, OBM faces a unique challenge: when an online query with large bid arrives, there may be no offline node that both connects to the query and has sufficient remaining budgets for it. This leads to missed matches for the queries with large bids, ultimately resulting in a low competitive ratio. To formally show the inherent difficulty of OBM without the small-bid and FLM assumptions, we present an upper bound on the competitive ratio for any deterministic online algorithms in the following proposition.

**Proposition 4.1.** *For OBM without small-bid or FLM assumptions, the competitive ratio of any deterministic online algorithm is upper bounded by $1 - \kappa$ for $\kappa \in (0,1]$. Specifically, the competitive ratio for any deterministic algorithm is zero when $\kappa = 1$ without the FLM assumption.*

The proof of the upper bound is deferred to Appendix A.1. The key ingredients of the proof is given below. The best competitive ratio for any deterministic algorithm is $\max_\pi \min_{G \in \mathcal{G}} CR(\pi, G)$ which

---

[1] In this paper, two graphs with different orders of online nodes are considered as two different graphs.

---

**Algorithm 1** Meta Algorithm (`MetaAd`)

---

**Require:** The function $\phi : [0, 1] \rightarrow [0, 1]$
1: **Initialization:** $\forall u \in \mathcal{U}$, the remaining budget $b_{u,0} = B_u$.
2: **for** $t$=1 to $V$, a new vertex $t \in \mathcal{V}$ arrives **do**
3:      For $u \in \mathcal{U}$, set $s_{u,t} = w_{u,t}\phi(\frac{b_{u,t-1}}{B_u})$ if $b_{u,t-1} - w_{u,t} \geq 0$, and $s_{u,t} = 0$ otherwise.
4:      **if** $\forall u \in \mathcal{U}, s_{u,t} = 0$ **then**
5:          Skip the online arrival $t$ ($x_t = $ null).
6:      **else**
7:          Select $x_t = \arg\max_{u \in \mathcal{U}} s_{u,t}$.
8:      **end if**
9:      Update budget: If $x_t \neq$ null, $b_{x_t,t} = b_{x_t,t-1} - w_{x_t,t}$; and $\forall u \neq x_t, b_{u,t} = b_{u,t-1}$.
10: **end for**

---

is no larger than $\max_\pi \min_{G \in \mathcal{G}'} CR(\pi, G)$ where $\mathcal{G}' \subset \mathcal{G}$. Thus, we can prove the upper bound by constructing a subset $\mathcal{G}'$ with difficult instances and deriving the best competitive ratio among the deterministic algorithms for this subset $\mathcal{G}'$. In our constructed $\mathcal{G}'$, each example has one offline node and the bid values for the first $V - 1$ rounds sum up to $1 - \kappa + \epsilon$ with $\epsilon > 0$ being infinitely small. We let $\mathcal{G}' = \mathcal{G}'_1 \bigcup \mathcal{G}'_2$ where we have $w_{u,V} = \kappa B_u$ for examples in $\mathcal{G}'_1$, and $w_{u,V} = 0$ for examples in $\mathcal{G}'_2$. The instances in $\mathcal{G}'$ illustrate the dilemma between matching a query for immediate reward or saving the budget for future matches. If an algorithm chooses to match all the queries in the first $V - 1$ rounds, it can lose a bid of $\kappa B_u$ for instances in $\mathcal{G}'_1$ because there is no sufficient budget to match the final query. Conversely, if an algorithm chooses to skip some queries in the first $V - 1$ rounds to save the budget, it can lose a bid of $\kappa B_u$ for the instances in $\mathcal{G}'_2$ because matching the final query of instances in $\mathcal{G}'_2$ earns zero bid. For these difficult instances in $\mathcal{G}'$, we formally derive the largest reward ratio of a deterministic algorithm to the offline optimal one which is the upper bound of the competitive ratio.

The upper bound of the competitive ratio $1 - \kappa$ shows that OBM becomes more difficult when the bid-budget ratio $\kappa$ gets larger. Intuitively, since the bid value is not fractional for each matching, given a larger $\kappa \in [0, 1]$, it is more likely for offline nodes to have insufficient budgets (i.e., unable to be matched to a query with a large bid value). Skipping a query to save budget is also risky because it can happen that the following queries have no positive bid. The FLM assumption can alleviate this difficulty because the remaining budget can be fully spent even if it is insufficient. A greedy algorithm can achieve a competitive ratio of $1/2$ for general bids with FLM [23]. By contrast, the upper bound of the competitive ratio *without* FLM cannot reach $1/2$ when the bid-budget ratio $\kappa$ is larger than $1/2$. There is even no non-zero competitive ratio when $\kappa = 1$. These observations reveal that without FLM, OBM becomes more difficult. The analysis without FLM assumption will help us to understand OBM better.

## 4.2 Meta Algorithm Design

We now present `MetaAd` in Algorithm 1, which is a meta online algorithm that reduces to many concrete algorithms with provable competitive ratios for OBM in the absence of small-bid and FLM assumptions.

`MetaAd` relies on a general discounting function $\phi : [0, 1] \rightarrow [0, 1]$. Given a new query $t$, `MetaAd` uses $\phi$ to score each offline node by discounting its bid value in Line 3, and selects the node with the largest score $s_{u,t}$ from Line 4 to Line 7. An offline node is scored zero if it has insufficient budget for the query $t$ ($b_{u,t-1} - w_{u,t} < 0$) or it has zero bid for the query $t$ ($w_{u,t} = 0$). If all the offline nodes are scored zero, the algorithm skips this query $t$.

The scoring in `MetaAd` reflects a balance between selecting an offline node with a large bid and saving budget for future. To select offline nodes with large bids, the score scales with the bid value $w_{u,t}$. Simultaneously, an increasing function $\phi$ maps the normalized remaining budget $\frac{b_{u,t-1}}{B_u}$ to a discounting value within $[0, 1]$. If an offline node $u$ has less remaining budget, a smaller discounting value is obtained to encourage conserving budget for $u$.

---

**Algorithm 2** Dual Construction

---

**Require:** The function $\phi : [0,1] \rightarrow [0,1]$, $\varphi(x) = 1 - \phi(1-x)$, $\rho \geq 1$.
1: **Initialization:** $\forall u \in \mathcal{U}$, $\alpha_{u,0} = 0$, $b_{u,0} = B_u$, $\mathcal{U}^\circ = \emptyset$, and $\forall t \in [V]$, $\beta_t = 0$.
2: **for** $t$=1 to $V$, a new vertex $t \in \mathcal{V}$ arrives **do**
3:  Append $\{u \mid b_{u,t-1} - w_{u,t} < 0\}$ into $\mathcal{U}^\circ$.
4:  Score $s_{u,t}$, select $x_t$ for the arrival $t$, and update budget $b_{u,t}$ by Algorithm 1.
5:  Set the dual variable $\beta_t = s_{x_t,t}$, and set the dual variable $\alpha_{u,t} = \varphi(\frac{c_{u,t}}{B_u})$.
6: **end for**
7: For $u \notin \mathcal{U}^\circ$, set $\alpha_u = \alpha_{u,V}$; and for $u \in \mathcal{U}^\circ$, set $\alpha_u = 1$.

---

### 4.3 Competitive analysis

Given any monotonically increasing function $\phi : [0,1] \rightarrow [0,1]$ in Algorithm 1, we can get a concrete algorithm for OBM. For different bid-budget ratio $\kappa$, the competitive ratio of `MetaAd` is given in the main theorem below.

**Theorem 4.2.** *If the function $\phi : [0,1] \rightarrow [0,1]$ in Algorithm 1 satisfies that given an integer $n \geq 1$, $\forall i \leq n$, $\varphi^{(i)}(x) > 0$ where $\varphi(x) = 1 - \phi(1-x)$, the competitive ratio of Algorithm 1 is*

$$\eta(\kappa) = \frac{1}{1 + \kappa^{n+1}R + \max_{y \in [0,1]} \Delta(y) + \frac{\phi(\kappa)}{1-\kappa}},$$

*where $R$ is the Lipschitz constant of $\varphi^{(n)}(x)$ if $\varphi^{(n)}(x)$ is not monotonically decreasing, and $R = 0$ otherwise. Additionally, $\Delta(y) = \frac{\varphi(y)}{y} - \frac{1}{y}\int_{x=0}^{y} \varphi(x)dx + \frac{1}{y}\sum_{i=1}^{n} \kappa^i \left(\varphi^{(i-1)}(y) - \varphi^{(i-1)}(0)\right)$.*

By Theorem 4.2, we can easily get a competitive algorithm for OBM with any bid-budget ratio $\kappa$ by choosing a function $\phi$. The only requirement is that the function $\phi$ is a monotonically increasing function. In the next section, we will give some concrete examples of competitive algorithms by assigning $\phi$ with different function classes.

We defer the complete proof of Theorem 4.2 to Appendix A.2. The analysis is based on the fundamental conditions in Lemma 1 that guarantee the competitive ratio and presents new challenges due to the absence of the small-bid and FLM assumptions.

**Lemma 1** (Conditions for competitive ratio). *An online algorithm achieves a competitive ratio of $\eta \in [0,1]$ if it selects a series of feasible actions $\{x_1, \ldots, x_V\}$ and there exist dual variables $\{\beta_1, \cdots, \beta_V\}$, $\{\alpha_1, \cdots, \alpha_U\}$ such that*

- *(**Dual feasibility**) $\forall u \in \mathcal{U}, t \in [V], \beta_t \geq w_{u,t}(1 - \alpha_u)$*

- *(**Primal-Dual Ratio**) $P \geq \eta \cdot D$, where $P = \sum_{t=1}^{V} w_{x_t,t}$ and $D = \sum_{u \in \mathcal{U}} B_u \alpha_u + \sum_{t=1}^{V} \beta_t$.*

Without the small-bid and FLM assumptions, the competitive analysis presents the following new challenges to satisfy the conditions in Lemma 1:
• **Dual construction for general bids.** When an offline node has an insufficient budget to match a query, the remaining budget is almost zero for the small-bid setting, but it can be large and uncertain without the small-bid assumption. This introduces a new challenge to construct dual variables that satisfy the dual feasibility due to budget insufficiency.

To address this challenge, we present a new dual construction in Algorithm 2 where dual variables are determined based on the remaining budget and adjusted at the end of the algorithm. The constructed dual variables satisfy the dual feasibility in Lemma 1, as explained below. We define $\beta_t$ as the score of selected offline node $u$. For any $u$ with sufficient remaining budget ($b_{u,t-1} \geq w_{u,t}$), we have $\beta_t \geq s_{u,t} = w_{u,t}(1 - \varphi(\frac{c_{u,t-1}}{B_u}))$. By choosing $\alpha_{u,t} = \varphi(\frac{c_{u,t}}{B_u})$, the dual feasibility in Lemma 1 is satisfied for $t$ and $u$ with sufficient budget ($b_{u,t-1} \geq w_{u,t}$) since by an increasing function $\varphi$, it holds that $\alpha_u \geq \alpha_{u,t}$ for any $t \in [T]$.

Different from the small-bid setting, we need to adjust the dual variables at the end of the dual construction (Line 7 in Algorithm 2) to satisfy dual feasibility. We set $\alpha_u = 1$ for any $u$ with insufficient budget ($b_{u,t-1} < w_{u,t}$) at the end of the dual construction. This ensures that the dual feasibility is always satisfied without FLM.

- **Guarantee the primal-dual ratio.** The challenges in guaranteeing the primal-dual ratio in Lemma 1 come from the unspecified discounting function $\phi$ and the absence of the small-bid and FLM assumptions. To solve this challenge, we derive a condition to satisfy the primal dual ratio $P_t \geq \frac{1}{\gamma} D_t$ for any round $t$ where $\gamma \geq 1$, $P_t = \sum_{i=1}^{t} w_{x_i,i}$ is the cumulative primal reward and $D_t = \sum_{u \in \mathcal{U}} B_u \alpha_{u,t} + \sum_{i=1}^{t} \beta_t$ is the cumulative dual. Thus, we prove that the primal dual ratio $P_t \geq \frac{1}{\gamma} D_t$ is satisfied for any $t \in [T]$ if for any $y \in [0, 1]$ it holds that,

$$\varphi(y) - \int_{x=0}^{y} \varphi(x)dx + \sum_{i=1}^{n} \kappa^i \varphi^{(i-1)}(y) + (\kappa^{n+1} R - \gamma + 1)y \leq \sum_{i=1}^{n} \kappa^i \varphi^{(i-1)}(0), \qquad (3)$$

where $\varphi(x) = 1 - \phi(1 - x)$ and $\phi$ is the discounting function. Given that the dual increase due to the final dual adjustment (Line 7 in Algorithm 2) is bounded by $\frac{\phi(\kappa)}{1-\kappa} \cdot P$, we can bound the final primal-dual ratio as $P \geq \frac{1}{\gamma + \frac{\phi(\kappa)}{1-\kappa}} D$. This leads to a competitive ratio of $\frac{1}{\gamma + \frac{\phi(\kappa)}{1-\kappa}}$. Given any discounting function $\phi$, we can solve for $\gamma$ that satisfies the condition in (3), thereby obtaining the competitive ratio of `MetaAd`.

### 4.4 Competitive Algorithm Examples

In this section, we assign $\phi$ with different functions to get concrete algorithms and competitive ratios.

#### 4.4.1 Small Bid

We first verify that `MetaAd` reduces to the optimal algorithm for small-bid setting ($\kappa \to 0$) [24, 23].

**Corollary 4.2.1.** *By choosing $\phi(x) = \frac{e - e^{1-x}}{e-1}$, `MetaAd` reduces to the algorithm in [24] and achieves the optimal competitive ratio of $1 - \frac{1}{e}$ for small-bid setting ($\kappa \to 0$).*

Corollary 4.2.1 shows that the competitive ratio in Theorem 4.2 is consistent with the classical results for small bids. Interestingly, our analysis shows how the optimal $\phi$ is obtained which is explained as follows. By solving (3) with "=" and $\kappa = 0$, we get

$$\varphi(x) = (\gamma - 1)e^x + 1 - \gamma, \quad \phi(x) = \gamma - (\gamma - 1)e^{1-x}, \qquad (4)$$

where $\gamma \leq \frac{e}{e-1}$ to make sure $\phi(x) \geq 0$ for $x \in [0, 1]$. By Theorem 4.2, we get the competitive ratio as $\lim_{\kappa \to 0} \frac{1}{\gamma + \frac{\rho\phi(\kappa)}{1-\kappa}} = \frac{1}{(2-e)\gamma + e}$. By optimally choosing $\gamma = \frac{e}{e-1}$, we get the optimal competitive ratio as $1 - \frac{1}{e}$.

#### 4.4.2 Exponential Function Class

Next, we consider an exponential function class $\varphi(x) = C_1 e^{\theta x} + C_2$ with $0 \leq \theta \leq 1$. To ensure $\varphi(x)$ is an increasing function, we choose $C_1 \geq 0$. Also, we choose $C_2 = -C_1$ to simplify the expression of the competitive ratio. We can observe that $\varphi(x)$ has positive $n$-th derivative for any $n \geq 1$. Thus, we choose $n = \infty$ in Theorem 4.2 to eliminate the term $\kappa^{n+1} R$. By substituting $\varphi(x)$ into $\eta(\kappa)$ in Theorem 4.2, we get the corollary below.

**Corollary 4.2.2.** *If we assign $\varphi(x) = Ce^{\theta x} - C$ with $C \geq 0$ and $0 \leq \theta \leq 1$ in `MetaAd` in Algorithm 1, we get the competitive ratio as*

$$\eta(\kappa) = \begin{cases} \frac{1}{1+C+C(1-\frac{1}{\theta}+\frac{\kappa}{1-\kappa\theta})(e^\theta-1)+\frac{1+C-Ce^{\theta(1-\kappa)}}{1-\kappa}}, & 1 - \frac{1}{\theta} + \frac{\kappa}{1-\kappa\theta} \geq 0 \\ \frac{1}{1+C+C(1-\frac{1}{\theta}+\frac{\kappa}{1-\kappa\theta})\theta+\frac{1+C-Ce^{\theta(1-\kappa)}}{1-\kappa}}, & 1 - \frac{1}{\theta} + \frac{\kappa}{1-\kappa\theta} < 0 \end{cases} \qquad (5)$$

We numerically solve the optimal $\eta(\kappa)$ for each $\kappa \in [0, 1]$ by adjusting the parameters $\theta$ and $C$ and show the results in Figure 1. We observe that `MetaAd` achieves a non-zero competitive ratio for $\kappa \in [0, 1)$. The competitive ratio for $\kappa = 0$ is the optimal competitive ratio of $1 - \frac{1}{e}$ for small-bid setting. The competitive ratio monotonically decreases with $\kappa$. This coincides with the intuition that when $\kappa$ gets larger, it is more likely to trigger budget insufficiency and the problem becomes more challenging. Also, we can find that for a large enough $\kappa$, the competitive ratio of `MetaAd` with the exponential function is very close to the upper bound. However, there can exist other forms of exponential discounting function that can achieve higher competitive ratio.

Interestingly, the optimal choices of the exponential function $\varphi(x)$ for different $\kappa$ reveal the insights into designing deterministic algorithms for OBM. When $\kappa$ is less than a critical point $\bar{\kappa} \approx 0.26$, the optimal choice of the exponential function is $\varphi(x) = \frac{1-e^{\theta x}}{1-e^{\theta}}$ and the optimal choice of $\theta$ decreases with $\kappa \in [0, \bar{\kappa}]$. In this range, as $\kappa$ becomes larger in $[0, \bar{\kappa}]$, the discounting $\phi(\frac{b_{u,t-1}}{B_u}) = 1 - \varphi(1 - \frac{b_{u,t-1}}{B_u})$ becomes smaller, indicating a more conservative approach to budget usage in preparation for potentially high future bids. However, as $\kappa$ gets larger than $\bar{\kappa}$, the optimal choice of the discounting function becomes $\phi(x) = 1$ with $C = 0$, yielding a greedy algorithm. This suggests that for large enough $\kappa$, the algorithm benefits more by matching a node with a large bid immediately than by conserving more budget for future.

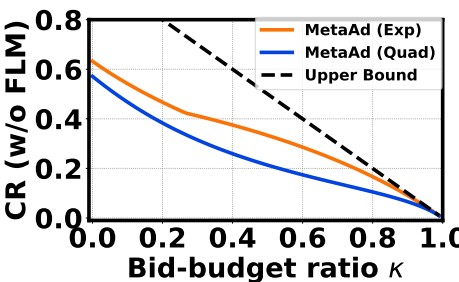

Figure 1: Competitive ratio without FLM. `MetaAd` (Exp) represents the `MetaAd` with $\varphi(x) = C(e^{\theta x} - 1)$ and `MetaAd` (Quad) represents the `MetaAd` with $\varphi(x) = Cx^2$.

### 4.4.3 Polynomial Function Class

In this section, we explore another function class to show that `MetaAd` is general enough to provide competitive algorithms given different discounting functions. We consider a function class of $n-$th polynomial function, i.e. $\varphi(x) = \sum_{j=0}^{n} C_j x^j$. We set $\sum_{j=1}^{n} C_j \leq 1$ to ensure $\varphi(1) \leq 1$ and set $C_0 = 0$ to simplify the competitive ratio. We summarize the competitive ratio of the polynomial function class and provide a concrete example for quadratic function in the next corollary.

**Corollary 4.2.3.** *If we assign $\varphi(x) = \sum_{j=1}^{n} C_j x^j$ with $\sum_{j=1}^{n} C_j \leq 1$ and $i$-th derivative $\varphi^{(i)}(x) \geq 0$ ($0 \leq i \leq n$), `MetaAd` achieves a competitive ratio as*

$$\eta(\kappa) = \frac{1}{1 + \max_{y \in [0,1]} \Delta(y) + \frac{1}{1-\kappa} - \sum_{j=1}^{n} C_j (1-\kappa)^{j-1}}, \quad (6)$$

*where $\Delta(y) = -\frac{C_n}{n+1} y^n + ((1 + \kappa) C_n - \frac{C_{n-1}}{n}) y^{n-1} + \sum_{j=0}^{n-2} ((1 + \kappa) C_{j+1} - \frac{C_j}{j+1} + \sum_{i=2}^{n-j} \kappa^i C_{i+j} \frac{(i+j)!}{(j+1)!}) y^j$. Specifically, given a quadratic example $\varphi(x) = Cx^2$, by optimally choosing $C = 1$, we have*

$$\eta(\kappa) = (\frac{11}{4} \kappa^2 + \frac{5}{2} \kappa + \frac{3}{4} + \frac{1}{1-\kappa})^{-1}. \quad (7)$$

We numerically show the results of $\eta(\kappa)$ in Figure 1. We observe that `MetaAd` with a simple quadratic function $\varphi(x) = x^2$ can also achieve non-zero competitive ratio for $\kappa \in [0, 1)$. However, this competitive ratio is lower than the best competitive ratio achieved by the exponential function $\varphi(x) = C(e^{\theta x} - 1)$.

The examples of exponential functions and quadratic functions demonstrate the strength of `MetaAd` in providing competitive algorithms for OBM with general bids. While `MetaAd` provides the first framework to get non-zero competitive ratio for OBM with $\kappa \in [0, 1)$ (in the absence of the FLM assumption), it is interesting to explore other functions $\phi$ under the `MetaAd` framework with better competitive ratios.

### 4.5 Extension to OBM with FLM

While `MetaAd` is designed for the more challenging OBM *without* FLM, this section demonstrates that `MetaAd` can be extended to provide competitive algorithms for OBM with FLM.

Due to the space limitation, we defer the algorithm of `MetaAd` with FLM (Algorithm 3) and its analysis to Appendix B. Instead of scoring based on the true bid $w_{u,t}$, Algorithm 3 determines the scores based on a modified bid $\min\{w_{u,t}, b_{u,t-1}\}$. Based on the modified dual construction in Algorithm 4, we can get the competitive ratio for OBM with FLM in the next theorem.

**Theorem 4.3.** *If the function $\phi : [0, 1] \to [0, 1]$ in Algorithm 1 satisfies that given an integer $n \geq 2$, $\forall i \leq n - 1$, $\varphi^{(i)}(x) > 0$ and $R = \max_{x \in [0,1]} \varphi^{(n)}(x)$ where $\varphi(x) = 1 - \phi(1 - x)$, the competitive ratio of Algorithm 1 is*

$$\eta(\kappa) = \frac{1}{1 + \kappa^n R + \max_{y \in [0,1]} \Delta(y) + \phi(\kappa)},$$

*where $\Delta(y) = \frac{\varphi(y)}{y} - \frac{1}{y} \int_{x=0}^{y} \varphi(x)dx + \frac{1}{y} \sum_{i=1}^{n} \kappa^i \left( \varphi^{(i-1)}(y) - \varphi^{(i-1)}(0) \right)$.*

The competitive ratio with FLM in Theorem 4.3 differs from that in Theorem 4.2 only in the final terms of the denominators, which are $\phi(\kappa)$ and $\frac{\phi(\kappa)}{1-\kappa}$ respectively. Thus, the competitive ratio with FLM is always larger than that without FLM given the same values of $\kappa$ and $\phi$. This improvement arises because FLM allows for accepting partial bids when budgets are insufficient, thereby reducing the potential budget waste.

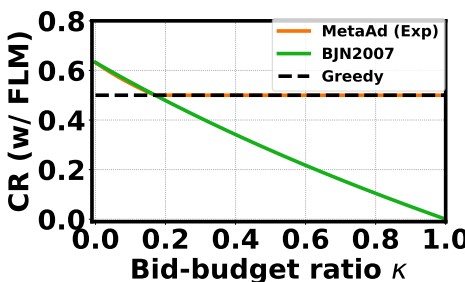

Figure 2: Competitive ratio with FLM. `MetaAd` (Exp) represents `MetaAd` with $\varphi(x) = C(e^{\theta x} - 1)$ and BJN2007 represents the algorithm in [4].

Similar as `MetaAd` without FLM, we assign an exponential function class $\varphi(x) = C(e^{\theta x} - 1)$ to get a concrete algorithm with competitive ratio in Corollary B.1.1. We numerically solve the optimal $\eta(\kappa)$ for each $\kappa \in [0, 1]$ by adjusting $\theta$ and $C$ and compare the results with an existing competitive algorithm BJN2007 [4] for FLM in Figure 2. As $\kappa \to 0$, both `MetaAd` and BJN2007 achieve the optimal competitive ratio $1 - 1/e$ in the small-bid setting. However, as $\kappa$ approaches 1, the competitive ratio of BJN2007 decreases to zero while `MetaAd`, reducing to a greedy algorithm, maintains a competitive ratio of $\frac{1}{2}$, the best known competitive ratio of the deterministic algorithms for the OBM with FLM.

## 5   Competitive Learning-Augmented Design

In this section, we demonstrate the application of our competitive analysis for designing learning-augmented algorithms which guarantee a competitive ratio of ML-based solutions for OBM.

Our competitive analysis directly motivates a learning-augmented algorithm for OBM called `LOBM`. The algorithm of `LOBM` and analysis are deferred to Appendix C. In `LOBM`, we apply a ML model which at each round takes the features of the arriving query and the offline nodes as inputs and gives the output $\tilde{z}_{u,t}$. Directly using $1 - \tilde{z}_{u,t}$ as a discounting value to set the score as $w_{u,t}(1 - \tilde{z}_{u,t})$ can result in arbitrarily bad worst-case performance for adversarial examples. To provide a competitive guarantee for OBM, `LOBM` projects the ML output $\tilde{z}_{u,t}$ into a competitive solution space $\mathcal{D}_{u,t}$ in (28) and obtains a projected value $z_{u,t}$. The score is then set as $w_{u,t}(1 - z_{u,t})$ based on the projected ML output $z_{u,t}$. The key design of the competitive solution space is motivated by the conditions in Lemma 1, which ensures that any $z$ value in $\mathcal{D}_{u,t}$ leads to the satisfactions of the dual feasibility and primal-dual ratio. The competitive solution space is based on the dual construction given the discounting function $\varphi(x) = \frac{e^{\theta x} - 1}{e^{\theta} - 1}$ where $\theta > 0$ in Algorithm 2. Importantly, we introduce a slackness parameter $\lambda \in [0, 1]$ in the design of $\mathcal{D}_{u,t}$ in (28). The parameter $\lambda$ controls the size of the competitive space $\mathcal{D}_{u,t}$ and further regulates the competitive ratio of LOBM. Given a smaller $\lambda$, we can get a larger competitive space $\mathcal{D}_{u,t}$, and so `LOBM` has more flexibility to exploit the benefits of ML predictions. However, a smaller $\lambda$ also leads to a smaller competitive ratio shown in the theorem below.

**Theorem 5.1.** *Given the maximum bid-budget ratio $\kappa \in [0, 1]$, $\theta > 0$, and the slackness parameter $\lambda \in [0, 1]$, with any ML predictions, `LOBM` in Algorithm 5 achieves a competitive ratio of*

$$\hat{\eta}(\kappa) = \frac{\lambda(1 - \frac{1}{e^{\theta}})}{1 + \lambda \left( \frac{1 - e^{-\theta \kappa}}{1 - \kappa} + (1 - \frac{1}{e^{\theta}}) \frac{1}{\theta} \left[ \frac{e^{\theta \kappa}}{\kappa} - \frac{1}{\kappa} - 1 \right]^+ \right)}. \tag{8}$$

|  | Algorithms w/o ML Predictions | | | ML-based Algorithms | | | |
|---|---|---|---|---|---|---|---|
|  | Greedy | PrimalDual | MetaAd | ML | LOBM-0.8 | LOBM-0.5 | LOBM-0.3 |
| **Worst-case** | 0.7941 | 0.8429 | **0.8524** | 0.7903 | **0.8538** | 0.8324 | 0.8113 |
| **Average** | 0.9329 | 0.9340 | **0.9344** | 0.9355 | **0.9372** | 0.9371 | 0.9343 |

Table 1: Worst-case and and average normalized reward on the MovieLens dataset. The best results among algorithms w/o ML predictions and the best results among ML-based algorithms are highlighted in bold font.

Theorem 5.1 shows that LOBM with a slackness parameter $\lambda \in [0, 1]$ can guarantee a competitiveness ratio of $\hat{\eta}(\kappa)$ regardless of the ML prediction quality. The parameter $\lambda \in [0, 1]$ determines the worst-case competitive ratio and the degree of flexibility to exploit the benefit of ML predictions. When $\lambda = 0$, there is no competitive ratio guarantee and LOBM reduces to a pure ML-based algorithm. This can also be seen from the inequalities in the competitive solution space (28), which are all satisfied automatically when $\lambda = 0$. On the other hand, when $\lambda = 1$, LOBM achieves the highest competitive ratio. When $\lambda$ increases from 0 to 1, the competitive solution space in (28) varies from whole solution space (with $\lambda = 0$) to the smallest competitive solution space (with $\lambda = 1$). Therefore, the choice of the slackness parameter $\lambda$ provides a trade-off between the competitive guarantee and the average performance by adjusting the level of exploiting the ML predictions.

# 6 Empirical Results

We evaluate the empirical performance of MetaAd and LOBM on two applications. The first application is Online Movie Matching where the platform needs to match each query to a movie advertiser with limited budget. The empirical results are obtained based on the MovieLens Dataset [12]. The main empirical results are shown in Table 1. We compare MetaAd with the algorithms without using ML (Greedy and PrimalDual introduced in Section D.1.1) and show that MetaAd achieves the best worst-case and average performance among them. Additionally, we validate that LOBM with a guarantee of competitive ratio in Theorem 5.1 achieves the best worst-case reward with a good average reward. Other empirical ablation studies can be found in Section D.1.2.

The second application is Online VM Placement introduced in Section 3. We generate the bipartite graphs with connections between physical servers and VMs by the Barabási–Albert method [3] and assign utility values according to the prices of Amazon EC2 compute-optimized instances [2]. We defer the empirical results and ablation studies to Appendix D.2.3.

# 7 Conclusion

In this paper, we consider a challenging setting for OBM without the FLM and small-bid assumption. First, we highlight the challenges by proving an upper bound on the competitive ratio for any deterministic algorithms in OBM. Then, we design the first meta algorithm MetaAd that achieves a provable competitive ratios parameterized by the maximum bid-budget ratio $\kappa \in [0, 1]$. We also extend LOBM under the additional FLM assumption. Additionally, based on the competitive analysis, we propose LOBM to take advantage of ML predictions to improve the performance with a competitive ratio guarantee, followed by its empirical validations.

**Limitations and Future Directions.** While we provide the first provable meta algorithms for OBM with general bids, determining the best choice of the discounting function $\phi$ remains an open question and an interesting problem for future exploration.

**Broader impacts.** By introducing a provable algorithm for OBM under more general settings, our work has the potential to advance the applications and motivate new algorithms. For applications like advertising, if large budget disparities among offline nodes exist, those with larger initial budgets could have a higher chance of being matched due to their smaller bid-to-budget ratios. This fairness issue, also observed in prior algorithms [23, 4, 24], warrants further investigation.

## Aknowledgement

Jianyi Yang, Pengfei Li and Shaolei Ren were supported in part by NSF grants CNS-2007115 and CCF-2324941. Adam Wierman was supported by NSF grants CCF-2326609, CNS-2146814, CPS-2136197, CNS-2106403, and NGSDI-2105648 as well as funding from the Resnick Sustainability Institute.

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

# A Proof of theorems in Section 4

## A.1 Proof of Proposition 4.1

*Proof.* Proposition 4.1 can be proved as follows. Denote $CR(\pi; G)$ as the competitive ratio of a deterministic algorithm $\pi$ on the graph instance $G$. The competitive ratio for any deterministic algorithm is $\max_\pi \min_{G \in \mathcal{G}} CR(\pi, G)$ which is no larger than $\max_\pi \min_{G \in \mathcal{G}'} CR(\pi, G)$ where $\mathcal{G}' \subset \mathcal{G}$. Thus, we can prove the upper bound of the competitive ratio by constructing an example subset $\mathcal{G}'$ and deriving the resulting competitive ratio for any deterministic algorithm. Specifically, an example subset $\mathcal{G}'$ is constructed as below.

**Example 1.** *Consider a setting with only one offline node and a total budget of $1$. The agent needs to decide whether or not to match an online node with a bid value $w_{u,t} \leq \kappa, \kappa \in (0, 1]$ to the offline node for $V \geq 2$ rounds. The bid values for the first $V - 1$ rounds are equivalent to $\omega$ and sum up to $1 - \kappa + \epsilon$ where $\epsilon$ is infinitely small, so we have $\omega \in (0, (1 - \kappa + \epsilon)/\lceil \frac{1-\kappa+\epsilon}{\kappa} \rceil]$. The bid value $w_{u,V}$ in the last round is either zero or $\kappa$ and is not known to the agent. Thus, the constructed example subset is composed of two smaller subsets, i.e. $\mathcal{G}' = \mathcal{G}'_1 + \mathcal{G}'_2$. In the example of the subset $\mathcal{G}'_1$, we have $w_{u,V} = \kappa$, and in the examples of the subset $\mathcal{G}'_2$, we have $w_{u,V} = 0$.*

The offline optimal solutions are different for $\mathcal{G}'_1$ and $\mathcal{G}'_2$ in Example 1. For $\mathcal{G}'_1$ with the last bid value as $w_{u,V} = \kappa$, the optimal solution is to skip one of the first $(V - 1)$ rounds. In this way, the last online node with bid $\kappa$ can be matched and the total reward is $1 + \epsilon - \omega$. For $\mathcal{G}'_2$ with the last bid value as $w_{u,V} = 0$, the optimal solution is to match all the online nodes for the first $V - 1$ rounds and obtain a total reward of $1 - \kappa + \epsilon$.

For the examples in $\mathcal{G}'$ in Example 1, the optimal online algorithm can be chosen from the following two. First, the algorithm can choose to match the online node to the offline node in all the first $(V - 1)$ rounds. This algorithm is optimal for $\mathcal{G}'_2$, but for $\mathcal{G}'_1$ with $w_{u,V} = \kappa$, the total reward is $1 - \kappa + \epsilon$ which is less than the offline optimal reward $1 + \epsilon - \omega$. Therefore, the competitive ratio of this algorithm in the worst case is $\lim_{\epsilon \to 0} \min_{\omega \in (0, (1-\kappa+\epsilon)/\lceil \frac{1-\kappa+\epsilon}{\kappa} \rceil]} \frac{1-\kappa+\epsilon}{1+\epsilon-\omega} \to 1 - \kappa$ when $\omega$ is infinitely small. Second, the algorithm can choose to skip one round in the first $V - 1$ rounds such that the last online node can be matched if it has a bid value of $\kappa$ in $\mathcal{G}'_1$. However, for $\mathcal{G}'_2$ with $w_{u,V} = 0$ and the total reward is $1 - \kappa + \epsilon - \omega$ which is less than the optimal reward as $1 - \kappa + \epsilon$. Thus, the competitive ratio of this algorithm is $\lim_{\epsilon \to 0} \min_{\omega \in (0, (1-\kappa+\epsilon)/(\lceil \frac{1-\kappa+\epsilon}{\kappa} \rceil])} \frac{1-\kappa+\epsilon-\omega}{1-\kappa+\epsilon} = 1 - \frac{1}{\lceil \frac{1-\kappa}{\kappa} \rceil}$ for $\kappa \in (0, 1)$. When $\kappa = 1$, the competitive ratio of this algorithm is $\min_{\omega \in (0, \epsilon)} \frac{1-\kappa+\epsilon-\omega}{1-\kappa+\epsilon} = \min_{\omega \in (0, \epsilon)} \frac{\epsilon-\omega}{\epsilon} = 0$. Therefore, the competitive ratio for any deterministic algorithm for $\mathcal{G}'$ is $\max_\pi \min_{G \in \mathcal{G}'} CR(\pi, G) = \max\{1 - \kappa, 1 - \frac{1}{\lceil \frac{1-\kappa}{\kappa} \rceil}\} = 1 - \kappa$ for $\kappa \in (0, 1)$, and $0$ for $\kappa = 1$.

Combining both cases of $\kappa \in (0, 1)$ and $\kappa = 1$, we get the upper bound of the competitive ratio for any deterministic algorithm for $\mathcal{G}'$ as $1 - \kappa$, which is also an upper bound of the competitive ratio of any deterministic algorithm for OBM. □

## A.2 Proof of Theorem 4.2

To prove Theorem A.2, we first prove Lemma 1.

**Proof of Lemma 1.**

*Proof.* The first condition guarantees that the dual variables are feasible. The second condition is to guarantee the competitive performance. Let $D^*$ and $P^*$ be the optimal dual and primal objectives. If the second condition is satisfied, then we have

$$P \geq \eta D \geq \eta D^* \geq \eta P^*, \tag{9}$$

where the second inequality holds since $D^*$ is the minimum dual objective, and the third inequality comes from weak duality. This completes the proof. □

**Proof of Theorem A.2**

*Proof.* To satisfy the primal-dual ratio $P_t \geq \frac{1}{\gamma} D_t$, we can get an inequality of $\alpha_{u,t}$ as below.

$$\sum_{u \in \mathcal{U}} B_u \alpha_{u,t} + \sum_{i=1}^{t} \beta_t \leq \gamma \cdot \sum_{i=1}^{t} w_{x_i,i}$$

$$\Leftrightarrow \sum_{u \in \mathcal{U}} B_u \alpha_{u,t} + \sum_{u \in \mathcal{U}} \sum_{i=1,x_i=u}^{t} w_{u,i}(1 - \varphi(\frac{c_{u,i-1}}{B_u})) \leq \gamma \cdot \sum_{u \in \mathcal{U}} c_{u,t}$$

$$\Leftarrow \forall u \in \mathcal{U}, B_u \alpha_{u,t} + \sum_{i=1,x_i=u}^{t} w_{u,i}(1 - \varphi(\frac{c_{u,i-1}}{B_u})) \leq \gamma \cdot c_{u,t}$$

$$\Leftrightarrow \forall u \in \mathcal{U}, \alpha_{u,t} \leq \sum_{i=1,x_i=u}^{t} \frac{w_{u,i}}{B_u} \varphi(\frac{c_{u,i-1}}{B_u}) + (\gamma - 1)\frac{c_{u,t}}{B_u},$$

(10)

where $c_{u,t} = \sum_{i=1,x_i=u}^{t} w_{x_i,i}$. Since $\alpha_{u,t} = \varphi(\frac{c_{u,t}}{B_u})$, we get the condition of $\varphi$ as below.

$$\varphi(\frac{c_{u,t}}{B_u}) \leq \sum_{i=1,x_i=u}^{t} \frac{w_{u,i}}{B_u} \varphi(\frac{c_{u,i-1}}{B_u}) + (\gamma - 1)\frac{c_{u,t}}{B_u}.$$

(11)

Further, since $\varphi$ is an increasing function, we have the following bound for the discrete sum.

$$\sum_{i=1,x_i=u}^{t} \frac{w_{u,i}}{B_u} \varphi(\frac{c_{u,i-1}}{B_u})$$

$$= \sum_{i=1,x_i=u}^{t} \frac{w_{u,i}}{B_u} \varphi(\frac{c_{u,i}}{B_u}) - \sum_{i=1,x_i=u}^{t} \frac{w_{u,i}}{B_u}(\varphi(\frac{c_{u,i}}{B_u}) - \varphi(\frac{c_{u,i-1}}{B_u}))$$

$$\geq \int_{x=0}^{\frac{c_{u,t}}{B_u}} \varphi(x)dx - \sum_{i=1,x_i=u}^{t} \frac{w_{u,i}}{B_u}(\varphi(\frac{c_{u,i}}{B_u}) - \varphi(\frac{c_{u,i-1}}{B_u}))$$

(12)

where the inequality holds by the integral inequality $\int_{x=0}^{y} \varphi(x)dx \leq \sum_{i=1}^{N} x_i \varphi(\sum_{j=1}^{i} x_j)$ with $y = \sum_{i=1}^{N} x_i$ for a positive increasing function $\varphi$. If $\varphi''(x) \leq 0$ for $x \in [0,1]$, we have

$$\sum_{i=1,x_i=u}^{t} \frac{w_{u,i}}{B_u} \varphi(\frac{c_{u,i-1}}{B_u})$$

$$\geq \int_{x=0}^{\frac{c_{u,t}}{B_u}} \varphi(x)dx - \sum_{i=1,x_i=u}^{t} (\frac{w_{u,i}}{B_u})^2 \varphi'(\frac{c_{u,i-1}}{B_u})$$

$$\geq \int_{x=0}^{\frac{c_{u,t}}{B_u}} \varphi(x)dx - \kappa \int_{x=0}^{\frac{c_{u,t}}{B_u}} \varphi'(x)dx$$

$$= \int_{x=0}^{\frac{c_{u,t}}{B_u}} \varphi(x)dx - (\kappa\varphi(\frac{c_{u,t}}{B_u}) - \kappa\varphi(0)),$$

(13)

where the first inequality holds since $\varphi''(x) \leq 0$ for $x \in [0,1]$, the second inequality holds by the bid bound $\kappa$ and another integral inequality $\int_{x=0}^{y} \varphi(x)dx \geq \sum_{i=1}^{N} x_i \varphi(\sum_{j=1}^{i-1} x_j)$ with $y = \sum_{i=1}^{N} x_i$.

If $0 < \varphi''(x) \le R$ for $x \in [0, 1]$, following Eqn. (12), we have

$$
\sum_{i=1, x_i=u}^{t} \frac{w_{u,i}}{B_u} \varphi(\frac{c_{u,i-1}}{B_u})
$$

$$
\ge \int_{x=0}^{\frac{c_{u,t}}{B_u}} \varphi(x)dx - \kappa \sum_{i=1, x_i=u}^{t} \frac{w_{u,i}}{B_u} \varphi'(\frac{c_{u,i-1}}{B_u}) - \kappa \sum_{i=1, x_i=u}^{t} \frac{w_{u,i}}{B_u} (\varphi'(\frac{c_{u,i}}{B_u}) - \varphi'(\frac{c_{u,i-1}}{B_u})),
$$

$$
\ge \int_{x=0}^{\frac{c_{u,t}}{B_u}} \varphi(x)dx - \kappa \int_{x=0}^{\frac{c_{u,t}}{B_u}} \varphi'(x)dx - \kappa \sum_{i=1, x_i=u}^{t} (\frac{w_{u,i}}{B_u})^2 R
$$

$$
= \int_{x=0}^{\frac{c_{u,t}}{B_u}} \varphi(x)dx - (\kappa\varphi(\frac{c_{u,t}}{B_u}) - \kappa\varphi(0)) - \kappa^2 R \frac{c_{u,t}}{B_u},
$$

(14)

where the second inequality holds by the integral inequality and $\varphi''(x) \le R$.

Therefore, we can extend to the case where $\varphi$ has $n-$the derivative. If $\varphi^{(i)}(x) > 0, \forall i \le n, \forall x \in [0, 1]$, then we have

$$
\sum_{i=1, x_i=u}^{t} \frac{w_{u,i}}{B_u} \varphi(\frac{c_{u,i-1}}{B_u})
$$

$$
\ge \int_{x=0}^{\frac{c_{u,t}}{B_u}} \varphi(x)dx - \sum_{i=1}^{n} \kappa^i \varphi^{(i-1)}(\frac{c_{u,t}}{B_u}) + \sum_{i=1}^{n} \kappa^i \varphi^{(i-1)}(0) - \kappa^{n+1} R \frac{c_{u,t}}{B_u},
$$

(15)

where $R$ is the Lipschitz constant of $\varphi^{(n)}(x)$ if $\varphi^{(n)}(x)$ is not monotonically decreasing and $R = 0$ otherwise.

Substituting (15) into (11), the condition becomes

$$
\varphi(\frac{c_{u,t}}{B_u}) \le (\gamma - 1)\frac{c_{u,t}}{B_u} + \int_{x=0}^{\frac{c_{u,t}}{B_u}} \varphi(x)dx - \sum_{i=1}^{n} \kappa^i \varphi^{(i-1)}(\frac{c_{u,t}}{B_u}) + \sum_{i=1}^{n} \kappa^i \varphi^{(i-1)}(0) - \kappa^{n+1} R \frac{c_{u,t}}{B_u}.
$$

(16)

Thus, a $\varphi$ function satisfies the primal dual ratio $P_t \ge \frac{1}{\gamma} D_t$ if it satisfies for any $y \in [0, 1]$,

$$
\varphi(y) - \int_{x=0}^{y} \varphi(x)dx + \sum_{i=1}^{n} \kappa^i \varphi^{(i-1)}(y) + (\rho\kappa^{n+1}R - \gamma + 1)y \le \rho \sum_{i=1}^{n} \kappa^i \varphi^{(i-1)}(0). \quad (17)
$$

Finally, we bound $\sum_{u \in \mathcal{U}^\circ} B_u \Lambda_u$ where $\Lambda_u = \alpha_u - \alpha_{u,V}$ is the dual increase at the end of the dual construction 2. $\Lambda_u > 0$ can hold for $u \in \mathcal{U}^\circ$ because $\alpha_u = 1$ for $u \in \mathcal{U}^\circ$ Thus, after the $V$ loops, we have for $u \in \mathcal{U}^\circ$

$$
B_u \Lambda_u = B_u (1 - \alpha_{u,V}) = B_u \left(1 - \varphi(\frac{c_{u,V}}{B_u})\right)
$$

$$
= B_u \phi(\frac{b_{u,V}}{B_u}) \le B_u \phi(\kappa)
$$

(18)

where the inequality holds since $b_{u,V} \le \kappa B_u$ for any $u \in \mathcal{U}^\circ$. Thus, we have

$$
\sum_{u \in \mathcal{U}^\circ} B_u \Lambda_u \le \sum_{u \in \mathcal{U}^\circ} B_u \phi(\kappa) \le \phi(\kappa) \cdot P \cdot \frac{\sum_{u \in \mathcal{U}^\circ} B_u}{\sum_{u \in \mathcal{U}^\circ} (1 - \kappa) B_u} = \frac{\phi(\kappa)}{1 - \kappa} \cdot P, \quad (19)
$$

where the second inequality holds because $P \ge \sum_{u \in \mathcal{U}^\circ} (1 - \kappa) B_u$ given that $c_{u,V} \ge (1 - \kappa) B_u$ for $u \in \mathcal{U}^\circ$.

Putting them together, we have $P \ge \frac{1}{\gamma}(D - \frac{\phi(\kappa)}{1-\kappa} \cdot P)$ which leads to the primal dual ratio as $P \ge \frac{1}{\gamma + \frac{\phi(\kappa)}{1-\kappa}} D$. To satisfy (17) for any $\varphi$, we can choose $\gamma \ge 1 + \kappa^{n+1}R + \frac{\varphi(y)}{y} - \frac{1}{y}\int_{x=0}^{y} \varphi(x)dx + \frac{1}{y}\sum_{i=1}^{n} \kappa^i \left(\varphi^{(i-1)}(y) - \varphi^{(i-1)}(0)\right)$ for any $y \in [0, 1]$. Combining with Lemma 1, we get the competitive ratio in Theorem 4.2. $\qquad \square$

---

**Algorithm 3** Meta Algorithm (`MetaAd` with FLM)

---

**Require:** The function $\phi : [0,1] \to [0,1]$

  **Initialization:** $\forall u \in \mathcal{U}$, the remaining budget $b_{u,0} = B_u$.

  **for** $t=1$ to $V$, a new vertex $t \in \mathcal{V}$ arrives **do**

    For $u \in \mathcal{U}$, set $s_{u,t} = w_{u,t}\phi(\frac{b_{u,t-1}}{B_u})$ if $b_{u,t-1} - w_{u,t} \geq 0$, and set $s_{u,t} = b_{u,t-1}\phi(\frac{b_{u,t-1}}{B_u})$, otherwise.

    **if** $\forall u \in \mathcal{U}, s_{u,t} = 0$ **then**

      Skip the online arrival $t$ ($x_t = \text{null}$).

    **else**

      Select $x_t = \arg\max_{u \in \mathcal{U}} s_{u,t}$.

    **end if**

    Update budget: If $x_t \neq \text{null}$, $b_{x_t,t} = b_{x_t,t-1} - w_{x_t,t}$; and $\forall u \neq x_t, b_{u,t} = b_{u,t-1}$.

  **end for**

---

## B   `MetaAd` for OBM with FLM

In this section, we extend `MetaAd` to the setting with FLM by allowing offline nodes with insufficient budgets to accept fractional bid values equal to their remaining budgets in their last matching. In other words, by matching an online arrival $t$ to an offline node $u \in \mathcal{U}$, the agent receives an actual reward of $\min\{w_{u,t}, b_{u,t-1}\}$, where $w_{u,t}$ is the bid value and $b_{u,t-1}$ is the available budget at the beginning of round $t$.

### B.1   Algorithm Design

Even under the FLM assumption, OBM with general bids is challenging because when an online node $t$ arrives, if the remaining budget $b_{u,t-1}$ of an offline node $u$ is smaller than the bid $w_{u,t}$, matching the arrival to this offline node can cause a reward loss of $w_{u,t} - b_{u,t-1}$, which increases with the bid value $w_{u,t}$. With FLM, the greedy algorithm (`Greedy`) can achieve a competitive ratio of 0.5 [23]. The competitive ratio achieved by a deterministic algorithm in [4] is $(1 - \kappa - \frac{1-\kappa}{(1+\kappa)^{1/\kappa}})$.

For OBM with FLM, we use a different meta algorithm as in Algorithm 3. When the remaining budget $b_{u,t-1}$ for an offline node $u$ is enough to accept arrival $t$ (i.e. $b_{u,t-1} \geq w_{u,t}$), the scoring strategy is the same as Algorithm 1 which sets the score as $s_{u,t} = w_{u,t}\phi(\frac{b_{u,t-1}}{B_u})$. Nonetheless, the scoring strategy is different from Algorithm 1 when the remaining budget $b_{u,t-1}$ of an offline node $u$ is insufficient for an online arrival $t$ (i.e. $b_{u,t-1} < w_{u,t}$). Without FLM, Algorithm 1 directly sets the score $s_{u,t}$ as zero to avoid the selection of offline node $u$. However, FLM allows matching an offline node $u$ to the online arrival $t$ and consuming all the remaining budget $b_{u,t-1}$ to obtain a reward of $b_{u,t-1}$. Thus, Algorithm 3 can be greedier and sets the score as $s_{u,t} = b_{u,t-1}\phi(\frac{b_{u,t-1}}{B_u})$ to balance the actual reward increment and the budget consumption. Given an increasing function $\phi$, the score increases with the remaining budget, and it is still possible to select an offline node with an insufficient but large enough remaining budget.

### B.2   Competitive Analysis

In this section, we provide the competitive ratio of Algorithm 3 for OBM with FLM and discuss the insights and analysis techniques. The competitive ratio is given in Theorem B.1 with its proof deferred to Appendix B.3.

To prove the competitive ratio of `MetaAd` for FLM, we still need to construct dual variables $\alpha_u, u \in \mathcal{U}$ $\beta_t, t \in [V]$, which assists with online matching with a provable competitive ratio. The dual construction procedure is given in Algorithm 4. At each round $t$, same as the dual construction without FLM in Algorithm 2, $\beta_t$ is set as the score of the selected offline node and $\alpha_{u,t}$ is set as $\varphi(\frac{c_{u,t}}{B_u})$. Different from Algorithm 2, $\alpha_u$ is set at the end of the algorithm as below to satisfy dual feasibility with the FLM assumption.

$$\alpha_u = \max\left\{\alpha_{u,V}, \left\{1 - \frac{b_{u,t-1}}{w_{u,t}}\phi(\frac{b_{u,t-1}}{B_u}), t \in \mathcal{T}^\circ\right\}\right\}, \tag{20}$$

---

**Algorithm 4** Dual Construction (with FLM)

---

**Require:** The function $\phi : [0, 1] \to [0, 1]$, $\varphi(x) = 1 - \phi(1 - x)$, $\rho \geq 1$.
   **Initialization:** $\forall u \in \mathcal{U}$, $\alpha_{u,0} = 0$, $b_{u,0} = B_u$, $\mathcal{U}^\circ = \emptyset$, $\mathcal{T}^\circ = \emptyset$, and $\forall t \in [V]$, $\beta_t = 0$.
   **for** $t$=1 to $V$, a new vertex $t \in \mathcal{V}$ arrives **do**
      If there exist $u$ such that $b_{u,t-1} - w_{u,t} < 0$, append $\{u \mid b_{u,t-1} - w_{u,t} < 0\}$ into $\mathcal{U}^\circ$ and append $t$ to $\mathcal{T}^\circ$.
      Score $s_{u,t}$, select $x_t$ for the arrival $t$, and update budget $b_{u,t}$ by Algorithm 3.
      Set the dual variable $\beta_t = s_{x_t,t}$, and set the dual variable $\alpha_{u,t} = \varphi(\frac{c_{u,t}}{B_u})$.
   **end for**
   For $u \notin \mathcal{U}^\circ$, set $\alpha_u = \alpha_{u,V}$; and for $u \in \mathcal{U}^\circ$, set $\alpha_u$ as Eqn. (20).

---

where $t \in \mathcal{T}^\circ$ is the round when budget insufficiency happens. This is to guarantee the dual feasibility $\beta_t \geq b_{u,t-1}(1 - \alpha_{u,t-1}) \geq w_{u,t}(1 - \alpha_u)$ when the offline node has an insufficient budget for an arrival. Note that the dual increment $\Lambda_u = \alpha_u - \alpha_{u,V}$ at the end of the Algorithm 4 can be less than the dual increment at the end of Algorithm 2, thus resulting in a better competitive ratio for OBM with FLM than without FLM.

With the constructed dual variables, the competitive ratio of `MetaAd` for OBM with FLM is given in the next theorem.

**Theorem B.1.** *If the function $\phi : [0, 1] \to [0, 1]$ in Algorithm 1 satisfies that given an integer $n \geq 2$, $\forall i \leq n-1$, $\varphi^{(i)}(x) > 0$ and $R = \max_{x \in [0,1]} \varphi^{(n)}(x)$ where $\varphi(x) = 1 - \phi(1-x)$, the competitive ratio of Algorithm 1 is*

$$\eta(\kappa) = \frac{1}{1 + \kappa^n R + \max_{y \in [0,1]} \Delta(y) + \phi(\kappa)},$$

*where* $\Delta(y) = \frac{\varphi(y)}{y} - \frac{1}{y} \int_{x=0}^{y} \varphi(x)dx + \frac{1}{y} \sum_{i=1}^{n} \kappa^i \left( \varphi^{(i-1)}(y) - \varphi^{(i-1)}(0) \right)$.

Given different $\varphi$, we can get concrete competitive algorithms for OBM with FLM. In this paper, we show the example of the competitive algorithm where $\varphi$ is from the exponential function class.

**Corollary B.1.1.** *If we assign $\varphi(x) = Ce^{\theta x} - C$ with $C \geq 0$ and $0 \leq \theta \leq 1$ in `MetaAd` in Algorithm 3, we get the competitive ratio as*

$$\eta(\kappa) = \begin{cases} \frac{1}{1 + C + C(1 - \frac{1}{\theta} + \frac{\kappa}{1-\kappa\theta})(e^\theta - 1) + 1 + C - Ce^{\theta(1-\kappa)}}, & 1 - \frac{1}{\theta} + \frac{\kappa}{1-\kappa\theta} \geq 0 \\ \frac{1}{1 + C + C(1 - \frac{1}{\theta} + \frac{\kappa}{1-\kappa\theta})\theta + 1 + C - Ce^{\theta(1-\kappa)}}, & 1 - \frac{1}{\theta} + \frac{\kappa}{1-\kappa\theta} < 0 \end{cases} \tag{21}$$

### B.3  Proof of Theorem B.1 (Theorem 4.3)

*Proof.* Denote an equivalent bid as $\bar{w}_{u,t} = \min\{w_{u,t}, b_{u,t-1}\}$. To guarantee the primal-dual ratio $P_t \geq \frac{1}{\gamma} D_t$, we get the condition as below.

$$\sum_{u \in \mathcal{U}} B_u \alpha_{u,t} + \sum_{i=1}^{t} \beta_t \leq \gamma \cdot \sum_{i=1}^{t} \bar{w}_{x_i,i}$$

$$\Leftrightarrow \sum_{u \in \mathcal{U}} B_u \alpha_{u,t} + \sum_{u \in \mathcal{U}} \sum_{i=1, x_i=u}^{t} \bar{w}_{u,i}(1 - \varphi(\frac{c_{u,i-1}}{B_u})) \leq \gamma \cdot \sum_{u \in \mathcal{U}} c_{u,t}$$

$$\Leftarrow \forall u \in \mathcal{U}, B_u \alpha_{u,t} + \sum_{i=1, x_i=u}^{t} \bar{w}_{u,i}(1 - \varphi(\frac{c_{u,i-1}}{B_u})) \leq \gamma \cdot c_{u,t} \tag{22}$$

$$\Leftrightarrow \forall u \in \mathcal{U}, \alpha_{u,t} \leq \sum_{i=1, x_i=u}^{t} \frac{\bar{w}_{u,i}}{B_u} \varphi(\frac{c_{u,i-1}}{B_u}) + (\gamma - 1)\frac{c_{u,t}}{B_u},$$

where $c_{u,t} = \sum_{i=1, x_i=u}^{t} \bar{w}_{x_i,i}$.

Since $\alpha_{u,t} = \varphi(\frac{c_{u,t}}{B_u})$, we get the condition of $\varphi$ as below.

$$\varphi(\frac{c_{u,t}}{B_u}) \leq \sum_{i=1,x_i=u}^{t} \frac{\bar{w}_{u,i}}{B_u}\varphi(\frac{c_{u,i-1}}{B_u}) + (\gamma-1)\frac{c_{u,t}}{B_u}. \tag{23}$$

Same as the setting with FLM, we can bound the discrete sum as below. If for $i \leq n-1$, $\varphi^{(i)}(x) > 0$, $x \in [0,1]$, then we have

$$\sum_{i=1,x_i=u}^{t} \frac{\bar{w}_{u,i}}{B_u}\varphi(\frac{c_{u,i-1}}{B_u})$$
$$\geq \int_{x=0}^{\frac{c_{u,t}}{B_u}} \varphi(x)dx - \sum_{i=1}^{n-1}\kappa^i\varphi^{(i-1)}(\frac{c_{u,t}}{B_u}) + \sum_{i=1}^{n-1}\kappa^i\varphi^{(i-1)}(0) - \kappa^n R\frac{c_{u,t}}{B_u}, \tag{24}$$

where $R$ is the Lipschitz constant of $\varphi^{(n)}(x)$ if $\varphi^{(n)}(x)$ is not monotonically decreasing and $R = 0$ otherwise. Thus, a $\varphi$ function satisfies the primal dual ratio $P_t \geq \frac{1}{\gamma}D_t$ if it holds for any $y \in [0,1]$ that

$$\varphi(y) - \int_{x=0}^{y}\varphi(x)dx + \sum_{i=1}^{n-1}\kappa^i\varphi^{(i-1)}(y) + (\kappa^n R - \gamma + \rho)y \leq \sum_{i=1}^{n-1}\kappa^i\varphi^{(i-1)}(0). \tag{25}$$

Finally, we bound $\sum_{u \in \mathcal{U}^\circ} B_u\Lambda_u$ where $\Lambda_u = \alpha_u - \alpha_{u,V}$. $\Lambda_u > 0$ can hold for $u \in \mathcal{U}^\circ$ because $\alpha_u = \max\left\{\alpha_{u,V}, \left\{1 - \frac{b_{u,t-1}}{w_{u,t}}\phi(\frac{b_{u,t-1}}{B_u}), t \in \mathcal{T}^\circ\right\}\right\}$ by Eqn. (20). Thus, for a request $t \in \mathcal{T}^\circ$, we have $b_{u,t-1} - w_{u,t} < 0$ and $b_{u,t-1} \leq \kappa B_u$. Since $\varphi$ is an increasing function, it holds that $\alpha_{u,V} \geq \alpha_{u,t}$ for $t \in [V]$. Thus, after the $V$ loops, we have for $u \in \mathcal{U}^\circ$

$$B_u\Lambda_u \leq B_u\left(1 - \frac{b_{u,t-1}}{w_{u,t}}\phi(\frac{b_{u,t-1}}{B_u}) - \alpha_{u,V}\right)$$
$$\leq B_u\left(1 - \frac{b_{u,t-1}}{w_{u,t}}\phi(\frac{b_{u,t-1}}{B_u}) - \alpha_{u,t-1}\right)$$
$$= B_u\left(1 - \frac{b_{u,t-1}}{w_{u,t}}\right)\left(1 - \varphi(\frac{b_{u,t-1}}{B_u})\right) \tag{26}$$
$$\leq (B_u - b_{u,t-1})\left(1 - \varphi(\frac{b_{u,t-1}}{B_u})\right)$$
$$\leq c_{u,V}\left(1 - \varphi(1 - \kappa)\right),$$

where the third inequality holds since $w_{u,t} \leq B_u$ and the last inequality holds since $B_u - b_{u,t-1} = c_{u,t-1} \leq c_{u,V}$. Thus, we have $\sum_{u \in \mathcal{U}^\circ} B_u\Lambda_u \leq P \cdot (1 - \varphi(1 - \kappa))$.

Putting them together, we have $P \geq \frac{1}{\gamma}(D - \phi(\kappa) \cdot P)$ which leads to the primal dual ratio as $P \geq \frac{1}{\gamma + \phi(\kappa)}D$. Since we have $\gamma \geq 1 + \kappa^{n+1}R + \frac{\varphi(y)}{y} - \frac{1}{y}\int_{x=0}^{y}\varphi(x)dx + \frac{1}{y}\sum_{i=1}^{n}\kappa^i\left(\varphi^{(i-1)}(y) - \varphi^{(i-1)}(0)\right)$ for any $y \in [0,1]$ to satisfy (25). Combining with Lemma 1, we get the competitive ratio in Theorem 4.3. $\qquad\square$

## C   Learning Augmented OBM

In this section, we exploit the competitive solution space in `MetaAd` and propose to augment `MetaAd` with ML predictions (called `LOBM`) to improve the average performance while still offering a guaranteed competitive ratio in the worst case.

### C.1   Algorithm Design

A main technical challenge for OBM is to estimate the discounting value that discounts the bid values by a factor for the matching decision at round $t$. Thus, an ML model can be potentially leveraged to

replace the manual design of score assignment. More specifically, we can utilize an ML model to predict a discounting factor $z_{u,t}$ and set the score as $s_{u,t} = w_{u,t}(1 - z_{u,t})$ for matching when the offline node $u$ has a sufficient budget for the online arrival $t$. That is, we incorporate ML predictions into `MetaAd` (i.e., `LOBM`) to explore alternative score assignment strategies that can outperform manual designs on average while still offering guaranteed competitiveness.

In learning-augmented online algorithms [31, 21], there exists an intrinsic trade-off between following ML predictions for average performance improvement and achieving better robustness in the worst case. Such trade-off between average and worst-case performances also exist in learning-augmented OBM ( `LOBM`). To better control the trade-off, we introduce a slackness parameter $\lambda \in [0, 1]$ to relax the competitiveness requirement while allowing `LOBM` to improve the average performance through ML-based scoring within the competitive solution space.

If we blindly use the ML prediction as the discounting factor for matching, competitive ratio cannot be satisfied due to the lack of worst-case competitiveness for ML predictions. Thus, to ensure that `LOBM` still offers guaranteed competitiveness, we consider a competitive solution space based on the conditions for dual variables specified in Lemma 1.

Based on the competitive analysis with the exponential function class, we design a learning-augmented algorithm (i.e., `LOBM`) in Algorithm 5, which leverages ML prediction $\tilde{z}_{u,t}$ to improve the average performance while guaranteeing the worst-case competitive ratio. The key idea is to construct dual variables as we solve the primal problem online and utilize the dual variables to calibrate the ML prediction $\tilde{z}_{u,t}$. In this way, the matching decisions by `LOBM` are guaranteed to be competitive in the worst case while utilizing the potential benefits of ML predictions.

We describe `LOBM` in Algorithm 5 as follows. At the beginning, we initialize the dual variables as zero. Whenever an online node arrives, the agent receives a ML prediction $\tilde{z}_{u,t}$ indicating the discounting factors for all the offline nodes $u \in \mathcal{U}$. Instead of directly using the ML prediction to set the scores and selecting the offline node, `LOBM` projects the ML prediction $\tilde{z}_{u,t}$ into the competitive space $\mathcal{D}_{u,t}$ by solving the following for all $u \in \mathcal{U}$,

$$z_{u,t} = \arg \min_{z \in \mathcal{D}_{u,t}} |z - \tilde{z}_{u,t}|, \tag{27}$$

which is a key step to ensure the competitive ratio. In order to better utilize the potential benefit of ML predictions, we use the projection operation in (27) to select the discounting factor $z_{u,t}$ out of competitive space $\mathcal{D}_{u,t}$, such that the selected $z_{u,t}$ is the closest to the ML prediction $\tilde{z}_{u,t}$. Then, the projected value $z_{u,t}$ is used to set the scores $s_{u,t}$ for offline nodes with sufficient budgets for the online arrival $t$, and the scores for offline nodes with insufficient budgets are set as zero and these offline nodes are appended to $\mathcal{U}^\circ$. The scores based on calibrated ML predictions are then used to select the offline node for matching.

As the key design to guarantee the competitive ratio, the competitive space $\mathcal{D}_{u,t}$ is based on the conditions for dual variables in Lemma 1 and the dual construction by the exponential function class (Corollary 4.2.2). The dual variables are constructed as follows. For the selected note $x_t$ and its score $s_{x_t,t}$, we update the dual variable $\alpha_{x_t,t}$ as $\alpha_{x_t,t-1} + \frac{w_{x_t,t} z_{x_t,t}}{\lambda \rho_\theta B_{x_t}} + \delta_{x_t,t}$, where $\rho_\theta = 1 - \frac{1}{e^\theta}$ with $\theta > 0$, $z_{x_t,t}$ is the discounting factor when setting the score of $x_t$ (Line 4 of Algorithm 5), and $\delta_{x_t,t} = \frac{\exp(\theta(1 - b_{x_t,t-1}/B_{x_t}))}{e^\theta - 1} \left[ \exp(\frac{\theta w_{x_t,t}}{B_{x_t}}) - 1 - \frac{w_{x_t,t}}{B_{x_t}} \right]$ is a variable relying on the bid value $w_{x_t,t}$ and the remaining budget $b_{x_t,t-1}$. For unselected offline nodes, we keep their dual variables $\alpha_{u,t}$ the same as $\alpha_{u,t-1}$ for $u \neq x_t$. The dual variable $\beta_t$ is set based on the score of the selected offline node, i.e. $\beta_t = \frac{1}{\lambda \rho_\theta} s_{x_t,t} = \frac{1}{\lambda \rho_\theta} w_{x_t,t}(1 - z_{x_t,t})$. By constructing dual variables in this way, when an action $x_t$ is selected, the primal objective $P_t = \sum_{\tau=1}^t w_{x_\tau, \tau}$ increases by $w_{x_t,t}$ and the dual objective $D_t = \sum_{\tau=1}^t B_{x_\tau} \alpha_{x_\tau, \tau} + \beta_\tau$ increases by $B_{x_t}(\alpha_{x_t,t} - \alpha_{x_{t-1},t-1}) + \beta_t = \frac{w_{x_t,t}}{\lambda \rho_\theta} + B_{x_t} \delta_{x_t,t}$. When an online arrival $t$ is skipped without any matching, both primal and dual objectives remain the same with no updates. Thus, we can always ensure that the primal objective and the dual objective satisfy $D_t = \frac{1}{\lambda \rho_\theta} P_t + \sum_{\tau=1}^t B_{x_\tau, \tau} \delta_{x_\tau, \tau}$ for each $t \in [T]$, leading to a bounded ratio of the primal objective to the dual objective at the end of each round. The parameter $\lambda$ can be used to adjust the bound of primal-dual ratio, leading to different competitive ratios.

Next, we need to ensure that conditions in Lemma 1 are always satisfied no matter which offline node $u \in \mathcal{U}$ is selected at each round $t$. Thus, we construct the competitive space $\mathcal{D}_{u,t}$ as below and

---

**Algorithm 5** Learning-Augmented OBM (LOBM, w/o FLM)

1: **Initialization:** $\forall u \in \mathcal{U}, b_{u,0} = B_u, \forall u \in \mathcal{U}, \alpha_{u,0} = 0, \beta_0, \cdots, \beta_V = 0.$
2: **for** $t=1$ to $V$, a new request $t \in \mathcal{V}$ arrives **do**
3:      Get the ML prediction $\tilde{z}_{u,t}, \forall u \in \mathcal{U}$
4:      Project $\tilde{z}_{u,t}$ into $\mathcal{D}_{u,t}$ in (28) and get $z_{u,t}, \forall u \in \mathcal{U}$.
5:      For all $u \in \mathcal{U}$, if $b_{u,t-1} - w_{u,t} \geq 0$, set score $s_{u,t} = w_{u,t}(1 - z_{u,t})$; otherwise, set score
        $s_{u,t} = 0$ and append $\{u \mid b_{u,t-1} - w_{u,t} < 0\}$ into $\mathcal{U}^{\circ}$.
6:      **if** $\forall u \in \mathcal{U}, s_{u,t} = 0$ **then**
7:         Skip the online arrival $t$ ($x_t =$ null).
8:      **else**
9:         Select $x_t = \arg\max_{u \in \mathcal{U}} s_{u,t}$.
10:     **end if**
11:     If $x_t \neq$ null, update budget $b_{x_t,t} = b_{x_t,t-1} - w_{x_t,t}$; and $\forall u \neq x_t, b_{u,t} = b_{u,t-1}$.
12:     Update dual variables $\beta_t = \frac{1}{\lambda \rho_\theta} s_{x_t,t}$. If $x_t \neq$ null, update $\alpha_{x_t,t} = \alpha_{x_t,t-1} +$
       $\frac{1}{\lambda \rho_\theta B_{x_t}} w_{x_t,t} z_{x_t,t} + \delta_{x_t,t}$.
13: **end for**
14: For $u \notin \mathcal{U}^{\circ}$, set $\alpha_u = \alpha_{u,V}$; and for $u \in \mathcal{U}^{\circ}$, set $\alpha_u = 1$.

---

project the ML predictions $\tilde{z}_{u,t}$ into $\mathcal{D}_{u,t}$ if they fall outside $\mathcal{D}_{u,t}$:

$$
\mathcal{D}_{u,t} = \left\{ z \geq 0 \mid \frac{1}{\lambda \rho_\theta} w_{u,t}(1 - z) \geq w_{u,t} - w_{u,t} \alpha_{u,t-1}, \right.
$$
$$
\left. \alpha_{u,t-1} + \frac{w_{u,t}}{\lambda \rho_\theta B_u} z + \delta_{u,t} \geq \frac{\exp(\theta(1 - (b_{u,t-1} - w_{u,t})/B_u)) - 1}{e^\theta - 1} \right\}. \tag{28}
$$

Since the dual variable $\beta_t$ is set as $\frac{1}{\lambda \rho_\theta} s_{x_t,t}$ after selecting the offline node $x_t$ with the highest $s_{u,t}$ and sufficient budgets, $\beta_t$ is no less than $\frac{1}{\lambda \rho_\theta} s_{u,t} = \frac{1}{\lambda \rho_\theta} w_{u,t}(1 - z_{u,t})$ for any $u \in \mathcal{U}$. Thus, as long as the first inequality in (28) is satisfied, we always have the dual feasibility $\beta_t \geq w_{u,t} - w_{u,t} \alpha_{u,t-1}$ in Lemma 1 if all the offline nodes have sufficient budgets for the arrival $t$. For the offline nodes with insufficient budgets for the online arrival $t$, we ensure the dual feasibility $\beta_t \geq w_{u,t} - w_{u,t} \alpha_{u,t-1}$ by setting their corresponding dual variable $\alpha_u$ as one after the matching process (Line 14 in Algorithm 5).

The second inequality in (28) sets a target for the increment of the dual variables $\alpha_{u,t}$, which forces the dual variable $\alpha_{u,t}$ to be larger when the remaining budget becomes less. In this way, the score of an offline node $u$ with fewer remaining budgets can be set lower to be conservative in consuming budgets. Also, since $\alpha_{u,t}$ is larger when the remaining budget is less, the second inequality in (28) guarantees a large enough dual variable $\alpha_{u,t}$ when $u$ has insufficient budget for an arrival $t$. This keeps the additional dual increment after the matching process (Line 14 in Algorithm 5) bounded and further guarantees a bounded primal-dual ratio in the second condition of Lemma 1.

As we discussed, if the discounting factor $z_{u,t}$ at each round satisfies the inequalities in (28), the primal variables and the constructed dual variables will satisfy the conditions in Lemma 1, and so a competitive ratio for OBM is guaranteed. The size of the set $\mathcal{D}_{u,t}$ is controlled by the hyper-parameter $\lambda$: with smaller $\lambda$, the size of $\mathcal{D}_{u,t}$ becomes larger because the inequalities are easier to be satisfied. We will rigorously prove that $\mathcal{D}_{u,t}$ is always non-empty given any $\lambda \in [0, 1]$ to enable feasible competitive solutions that guarantee the competitive ratio bound in (29) in the robustness analysis of LOBM in Section C.2.

**ML model training and inference**. Given any ML predictions, Algorithm 5 provides a guarantee for the competitive ratio. Nonetheless, the average performance $\mathbb{E}_{\mathcal{G}}[P(\pi, \mathcal{G})]$ depends on the ML model that yields the ML prediction. Here, we briefly discuss how to achieve high average performance by training the ML model in an environment that is aware of the design of Algorithm 5. Note first that the projection operation is differentiable while the discrete matching decision is not differentiable. Thus, we apply policy gradient to train the ML model. Once the ML model is trained offline, it can be applied online to provide $\tilde{z}_{u,t}$ as advice for scoring and matching by LOBM (Line 3 in Algorithm 5).

## C.2 Analysis

Now, we provide a robustness analysis of LOBM and formally show that LOBM always guarantees the competitive ratio.

A learning-augmented algorithm is robust if its competitive ratio is guaranteed for any problem instance given arbitrary ML predictions. We show that LOBM is robust in the sense that it offers competitive guarantees regardless of the quality of ML predictions.

**Theorem C.1.** *Given the maximum bid-budget ratio $\kappa \in [0,1]$ and the slackness parameter $\lambda \in [0,1]$, with any ML predictions, LOBM in Algorithm 5 achieves a competitive ratio of*

$$\hat{\eta}(\kappa) = \frac{\lambda(1 - \frac{1}{e^\theta})}{1 + \lambda\left(\frac{1-e^{-\theta\kappa}}{1-\kappa} + (1 - \frac{1}{e^\theta})\frac{1}{\theta}\left[\frac{e^{\theta\kappa}}{\kappa} - \frac{1}{\kappa} - 1\right]^+\right)}, \tag{29}$$

*where $[x]^+ = x$ if $x > 0$ and $[x]^+ = 0$ if $x \leq 0$.*

Theorem C.1 shows that LOBM can guarantee a competitiveness ratio of $\hat{\eta}(\kappa)$ regardless of the ML prediction quality for any slackness parameter $\lambda \in [0,1]$. The parameter $\lambda \in [0,1]$ determines the requirement for the worst-case competitive ratio and the flexibility to exploit the benefit of ML predictions. When $\lambda = 0$, there is no competitiveness requirement, the inequalities in the competitive space (28) always hold, and LOBM reduces to a pure ML-based algorithm with no competitive ratio guarantee. On the other hand, when $\lambda = 1$, LOBM achieves the highest competitive ratio. When $\lambda$ is flexibly chosen between the competitive solution space also varies from whole solution space (with $\lambda = 0$) to the smallest competitive solution space (with $\lambda = 1$) in (28). Thus, LOBM achieves a flexible trade-off between the competitive guarantee and average performance by varying the levels of trust in ML predictions.

## C.3 Proof of Theorem C.1 (Theorem 5.1 in the main text)

*Proof.* The sequence of dual variables is constructed by Algorithm 5 We prove three claims leading to Theorem C.1.

- The dual feasibility is satisfied, i.e. $\forall u \in \mathcal{U}, t \in [V], w_{u,t}\alpha_u + \beta_t \geq w_{u,t}$.

- The primal-dual ratio is guaranteed, i.e. $P \geq \eta D$.

- The solution of projection (28) always exists, i.e. the feasible set of (28) is not empty for each round.

First, we prove the feasibility of dual variables (The first condition in Lemma 1). If $\beta_t = 0$ for a round $t \in [T]$, we have either $w_{u,t} = 0$ or $w_{u,t} > 0$ and $\alpha_u = \alpha_{u,V} \geq 1$ holds for a slot $u \in \mathcal{U}$ by Line 14, so $\beta_t = 0 \geq w_{u,t}(1 - \alpha_u)$ holds for the dual construction. On the other hand, if $\beta_t > 0$ holds for round $t \in [T]$, then the score $s_{x_t,t}$ must be calculated based on the projected $z_{x_t,t}$. Thus, we have $\beta_t = \frac{1}{\lambda\rho_\theta}s_{x_t,t} \geq \frac{1}{\lambda\rho_\theta}s_{u,t} = \frac{1}{\lambda\rho_\theta}w_{u,t}(1 - z_{u,t}) \geq w_{u,t}(1 - \alpha_{u,t}) \geq w_{x_t,t}(1 - \alpha_u)$ where $\rho_\theta = \frac{e^\theta-1}{e^\theta}$ and the second inequality holds by the first inequality of the set (28). This proves the feasibility of the dual variables

Next, we prove the primal-dual ratio (the second condition in Lemma 1) is satisfied. At round $t$, if no vertex is selected for a vertex $t$, the primal objective $P$ and dual objective $D$ do not increase. Otherwise, the primal objective increases by $w_{x_t,t}$ and dual objective increases by $B_{x_t}(\alpha_{x_t,t} - \alpha_{x_{t-1},t-1}) + \beta_t = \frac{w_{x_t,t}}{\lambda\rho_\theta} + B_{x_t}\delta_{x_t,t}$ where $\delta_{x_t,t} = \frac{\exp(\theta(1-b_{x_t,t-1}/B_{x_t}))}{e^\theta-1}(\exp(\frac{\theta w_{x_t,t}}{B_{x_t}}) - 1 - \frac{w_{x_t,t}}{B_{x_t}})$. By Line 14 at the end of Algorithm 5, the dual variable $\alpha_{u,V}$ increases to $\alpha_u$ by $\Lambda_u = \alpha_u - \alpha_{u,V}$. Thus, the dual objective can be written as $D = \frac{1}{\lambda\rho_\theta}\sum_{t=1}^V w_{x_t,t} + \sum_{t=1}^V B_{x_t}\delta_{x_t,t} + \sum_{u\in\mathcal{U}} B_u\Lambda_u$, and further we have

$$P = \sum_{t=1}^V w_{x_t,t} \geq \lambda(1 - \frac{1}{e^\theta})\left(D - \sum_{u\in\mathcal{U}^\circ} B_u\Lambda_u - \sum_{t=1}^V B_{x_t}\delta_{x_t,t}\right). \tag{30}$$

To bound the right-hand-side, we have

$$\sum_{u\in\mathcal{U}^\circ} B_u \Lambda_u \leq \frac{e^\theta(1-e^{-\theta\kappa})}{e^\theta-1}\sum_{u\in\mathcal{U}^\circ} B_u \leq \frac{e^\theta(1-e^{-\theta\kappa})P}{e^\theta-1}\frac{\sum_{u\in\mathcal{U}^\circ}B_u}{P}$$

$$\leq \frac{e^\theta(1-e^{-\theta\kappa})P}{e^\theta-1}\frac{\sum_{u\in\mathcal{U}^\circ}B_u}{\sum_{u\in\mathcal{U}^\circ}(1-\kappa)B_u} = \frac{e^\theta}{e^\theta-1}\frac{1-e^{-\theta\kappa}}{(1-\kappa)}\cdot P, \tag{31}$$

For each $u\in\mathcal{U}$, we sum up $B_u\delta_{u,t}$ and get

$$\sum_{t=1,x_t=u}^{V} B_{x_t}\delta_{x_t,t} = \sum_{t=1,x_t=u}^{V} \frac{\exp(\theta(1-b_{u,t-1}/B_u))}{e^\theta-1}(B_u\exp(\frac{\theta w_{u,t}}{B_u})-B_u-w_{u,t})$$

$$= \sum_{t=1,x_t=u}^{V} \frac{\exp(\theta(1-b_{u,t-1}/B_u))}{e^\theta-1}\cdot w_{u,t}\cdot\left(\frac{B_u}{w_{u,t}}\exp(\frac{\theta w_{u,t}}{B_u})-\frac{B_u}{w_{u,t}}-1\right)$$

$$\leq B_u\sum_{t=1,x_t=u}^{V} \frac{\exp(\theta(1-b_{u,t-1}/B_u))}{e^\theta-1}\cdot\frac{w_{u,t}}{B_u}\cdot\left[\frac{e^{\theta\kappa}}{\kappa}-\frac{1}{\kappa}-1\right]^+$$

$$\leq B_u\frac{\exp(\theta\frac{c_{u,V}}{B_u})-1}{\theta(e^\theta-1)}\cdot\left[\frac{e^{\theta\kappa}}{\kappa}-\frac{1}{\kappa}-1\right]^+$$

$$\leq c_{u,V}\cdot\frac{1}{\theta}\cdot\left[\frac{e^{\theta\kappa}}{\kappa}-\frac{1}{\kappa}-1\right]^+, \tag{32}$$

where the first inequality holds because $\frac{e^{\theta x}}{x}-\frac{1}{x}-1$ is an increasing function for $x\in[0,1],\theta\in[0,1]$, the second inequality holds since $\sum_{t=1,x_t=u}^{V}\exp(\theta(1-\frac{b_{u,t-1}}{B_u}))\cdot\frac{w_{u,t}}{B_u} = \sum_{t=1,x_t=u}^{V}\exp(\theta\frac{c_{u,t}}{B_u})\cdot\frac{w_{u,t}}{B_u} \leq \int_{x=0}^{\frac{c_{u,V}}{B_u}}\exp(\theta x)\mathrm{d}x = \frac{1}{\theta}(\exp(\theta\frac{c_{u,V}}{B_u})-1)$, and the last inequality holds because $B_u\frac{\exp(\theta\frac{c_{u,V}}{B_u})-1}{\theta(e^\theta-1)} = c_{u,V}\cdot\frac{\exp(\theta\frac{c_{u,V}}{B_u})-1}{\theta(e^\theta-1)\frac{c_{u,V}}{B_u}} \leq c_{u,V}\cdot\frac{1}{\theta}$.

By summing up all bidders in $\mathcal{U}$, we get

$$\sum_{t=1}^{V} B_{x_t}\delta_t = \sum_{u\in\mathcal{U}}\sum_{t=1,x_t=u}^{V} B_{x_t}\delta_{x_t,t} \leq P\cdot\frac{1}{\theta}\cdot\left[\frac{e^{\theta\kappa}}{\kappa}-\frac{1}{\kappa}-1\right]^+ \tag{33}$$

Continuing with inequality (30), we have

$$P \geq \lambda(1-\frac{1}{e^\theta})\left(D-\frac{e^\theta}{e^\theta-1}\frac{1-e^{-\theta\kappa}}{(1-\kappa)}\cdot P-\frac{1}{\theta}\cdot\left[\frac{e^{\theta\kappa}}{\kappa}-\frac{1}{\kappa}-1\right]^+\cdot P\right), \tag{34}$$

Thus, by moving terms, we have

$$P \geq \frac{\lambda(1-\frac{1}{e^\theta})D}{1+\lambda\left(\frac{1-e^{-\theta\kappa}}{1-\kappa}+(1-\frac{1}{e^\theta})\frac{1}{\theta}\left[\frac{e^{\theta\kappa}}{\kappa}-\frac{1}{\kappa}-1\right]^+\right)}. \tag{35}$$

This proves the second condition in Theorem 1.

Finally, we prove that the solution of the projection into (28) always exists, i.e. the feasible set of (28) is not empty for each round. To do this, we prove by induction that

$$z_{u,t}^\dagger = \frac{\lambda_1\rho_\theta\exp(\theta(1-b_{u,t-1}/B_u))}{(e^\theta-1)}, \forall\lambda_1\in[\lambda,1] \tag{36}$$

is always feasible for the set (28). For the first round, the initialized dual variable $\alpha_{u,0}=0$, the initialized remaining budget $b_{u,0}=B_u$, so we have

$$\alpha_{u,0}+\frac{w_{u,t}}{\lambda\rho_\theta B_u}z_{u,1}^\dagger+\delta_{u,1} \geq \frac{\exp(\theta(\frac{w_{u,t}}{B_u}))-1}{e^\theta-1}, \tag{37}$$

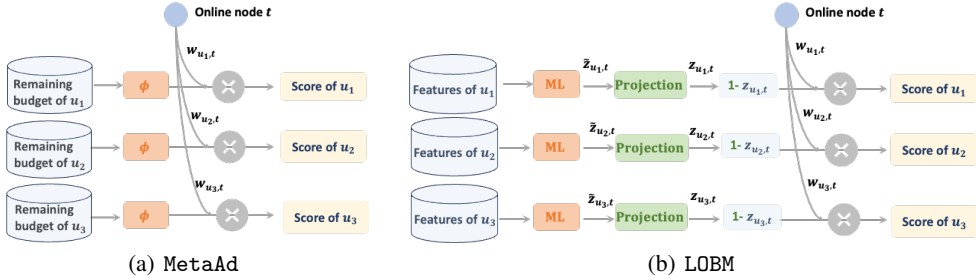

Figure 3: Illustration of the scoring strategies in `MetaAd` and `LOBM`. The example has 3 offline nodes $(u_1, u_2, u_3)$. The algorithms select the offline node with the largest score.

where the inequality holds because $\lambda_1 \geq \lambda$, so the second inequality of (28) holds for $t = 1$.

Also, it holds that

$$\frac{1}{\lambda \rho_\theta} w_{u,1} \left(1 - z_{u,1}^\dagger\right) = w_{u,1} \left(\frac{e^\theta}{\lambda(e^\theta - 1)} - \frac{\lambda_1}{\lambda(e^\theta - 1)}\right) \geq \frac{w_{u,1}}{\lambda} \geq w_{u,1} = w_{u,1}(1 - \alpha_{u,0}),$$
(38)

where the first inequality holds since $\lambda_1 \leq 1$ and the second inequality holds since $\lambda \leq 1$. Thus, we can prove the first inequality of (28) holds for initialization.

Then if the constraints are satisfied for the round before arrival $t$, then we have $\alpha_{u,t-1} \geq \frac{\exp(\theta(1 - b_{u,t-1}/B_u)) - 1}{e^\theta - 1}$, and thus we have

$$
\begin{aligned}
&\alpha_{u,t-1} + \frac{w_{u,t}}{\lambda \rho_\theta B_u} z_{u,t}^\dagger + \delta_{u,t} \\
&\geq \frac{\exp(\theta(1 - b_{u,t-1})/B_u)}{e^\theta - 1} \exp(\frac{\theta w_{u,t}}{B_u}) = \frac{\exp(\theta(1 - (b_{u,t-1} - w_{u,t})/B_u)}{e^\theta - 1}.
\end{aligned}
$$
(39)

where the inequality holds because $\lambda_1 \geq \lambda$. Thus the second inequality of (28) holds.

Then, since $\alpha_{u,t-1} \geq \frac{\exp(\theta(1 - b_{u,t-1}/B_u)) - 1}{e^\theta - 1}$, we have

$$
\begin{aligned}
&\frac{1}{\lambda \rho_\theta} w_{u,t} \left(1 - z_{u,t}^\dagger\right) \\
&= w_{u,t} \left(\frac{e^\theta}{\lambda(e^\theta - 1)} - \frac{\lambda_1 \exp(\theta(1 - \frac{b_{u,t-1}}{B_u}))}{\lambda(e^\theta - 1)}\right) \geq w_{u,t}(1 - \alpha_{u,t-1}),
\end{aligned}
$$
(40)

where the inequality holds since $\lambda \leq 1$ and $\lambda_1 \leq 1$. Thus we prove the first inequality of (28) holds. In conclusion, we can always find a feasible dual variable update $z_{u,t}^\dagger$ in the projection set (28). $\quad\square$

# D   Empirical Results

To complement the theoretical analysis, we validate the empirical benefits of proposed algorithms by conducting numerical experiments for an online movie matching application.

## D.1   Online Movie Matching

We evaluate the performances of our algorithms on the online movie matching application based on the MovieLens Dataset [12].

### D.1.1   Setup

In the application of online movie matching, each movie (i.e., an offline node) has a maximum budget set by advertisers. Once an online query arrives, the bid values of the query for all the movies are

| | Algorithms w/o ML Predictions | | | ML-based Algorithms | | | |
|---|---|---|---|---|---|---|---|
| | Greedy | PrimalDual | MetaAd | ML | LOBM-0.8 | LOBM-0.5 | LOBM-0.3 |
| **Worst-case** | 0.7941 | 0.8429 | **0.8524** | 0.7903 | **0.8538** | 0.8324 | 0.8113 |
| **Average** | 0.9329 | 0.9340 | **0.9344** | 0.9355 | **0.9372** | 0.9371 | 0.9343 |

Table 2: Worst-case and and average normalized reward on the MovieLens dataset. The best results among algorithms w/o ML predictions and the best results among ML-based algorithms are highlighted in bold font.

revealed to the matching platform agent. The platform agent needs to match a movie to each query, generating a reward equivalent to the bid value and consuming a budget of the bid value from the total budget of the matched movie. The bid value is determined by the relevance of the movie and the query. For example, if a movie is more relevant to the online query, there is a potentially higher value. The goal of the advertising platform is to maximize the total reward while satisfying the budget constraints of each movie. In this application, the platform agent does not allow a fractional fee for any matching. Thus, this matching problem is a OBM without FLM.

We run the online movie matching application based on a real dataset of MovieLens [12]. The MovieLens dataset provides data on the relevance of movies and users. We generate bipartite graphs, each with $U = 10$ offline nodes (movies) and $V = 100$ online nodes (queries/users) based on the MovieLens dataset. For each graph instance, we sample 10 movies uniformly without replacement and 100 users uniformly with replacement. A bid value scaled based on relevance is assigned to each edge between the offline node and the online node. The total budget for each offline node is sampled from a normal distribution with a mean of 1 and a standard deviation of 0.1, and the maximum bid value is 0.1 (i.e., $\kappa = 0.1$). We generate 10k, 1k, and 1k samples of graph instances based on the MovieLens dataset for training, validation and testing, respectively. To test the ML performance with out-of-distribution/adversarial examples, we also create examples by modifying $10\%$ examples in the testing dataset and randomly removing edges and/or rescaling the weights.

We compare our algorithms with the most common baselines for OBM as listed below.

- `OPT`: The offline optimal solution is obtained using Gurobi [11] for each graph instance.
- `Greedy`: The greedy algorithm [23] matches an online node to the available offline node that is connected to the node and has the highest bid value. `Greedy` has a strong empirical performance and is a special case of `MetaAd` with $\theta \to \infty$.
- `PrimalDual`: `PrimalDual` [23] calculates the scores of each bidder for each online node based on both the bid values and the remaining budgets, and then selects for an online node the available bidder with the highest score. It is a special case of `MetaAd` with $\theta \to 1$.
- `ML`: A policy-gradient algorithm that solves the OBM problem [1]. The inputs to the policy model are the available history information including the current bid value, the remaining budget of each offline node and the average matched bid value.

We evaluate the performances of `MetaAd` with the discounting function $\varphi(x) = \frac{e^{\theta x}-1}{e^\theta-1}$ and `LOBM` in Algorithm 5. The illustration of `MetaAd` for round $t$ is shown in Figure 3(a). `LOBM`-$\lambda$ is `LOBM` with the slackness parameter $\lambda$ in the competitive solution space (28). The illustration of `LOBM` for round $t$ is given in Figure 3(b). The The optimal parameter $\theta$ governing the level of conservativeness in `MetaAd` is tuned based on the validation dataset. We also evaluate the performance of `MetaAd` under different choices of $\theta$, and evaluate `LOBM` with ML predictions under different choices of the hyper-parameter $\lambda \in [0, 1]$ and use `LOBM`-$\lambda$ to represent `LOBM` with the hyper-parameter $\lambda$ in $\mathcal{D}_{u,t}$ in (28). For a fair comparison, we use the same neural architecture as `ML` in `LOBM`. The neural network has two layers, each with 200 hidden neurons. The neural networks are trained by Adam optimizer with a learning rate of $10^{-3}$ for 50 epochs. The training process on a laptop takes around 1 hour, while the inference process over each instance takes less than one second.

### D.1.2 Results

The empirical worst-case and average reward (normalized by the optimal reward) based on the MovieLens dataset are shown in Table 2. In this table, the parameter $\theta$ of `MetaAd` is 0.7 by default which is obtained by tuning on a validation dataset. We find that `MetaAd` can achieve a higher

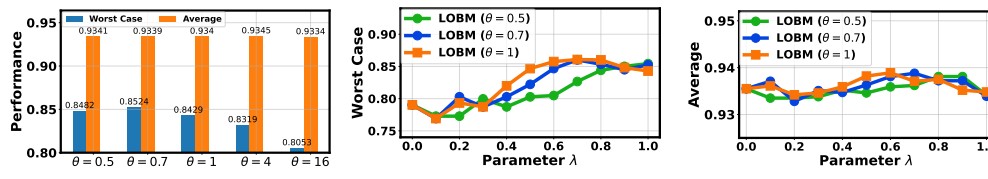

(a) Worst-case/average reward of MetaAd

(b) Worst-case reward of LOBM

(c) Average reward of LOBM

Figure 4: (a) Worst-case and average reward of MetaAd with different choices of $\theta$. (b) Worst-case reward of LOBM with different choices of $\theta$ and $\lambda$. (c)Average reward of LOBM with different choices of $\theta$ and $\lambda$.

worst-case reward ratio than alternative competitive algorithms without predictions (i.e., Greedy and PrimalDual). Through training, ML can achieve a higher average reward than competitive algorithms without predictions. However, due to the existence of out-of-distribution testing examples, ML has a lower worst-case reward ratio than competitive algorithms that have theoretical worst-case performance guarantees. LOBM can significantly improve the worst-case performance of ML. This is because the projection of ML predictions onto the competitive solution space in 28 corrects low-quality ML predictions. Interestingly, LOBM with $\lambda = 0.8$ achieves the best empirical worst-case and average performance, demonstrating the superiority of LOBM despite that its competitive ratio is lower than that of MetaAd. The high average performance of LOBM shows that LOBM can effectively utilize the benefits of good ML predictions to improve the average performance while offering guaranteed competitiveness. Importantly, when $\lambda \in [0, 1]$ decreases, the requirements for the worst-case performance are more relaxed, and hence LOBM achieves a higher average reward but a lower worst-case reward.

**The effects of $\theta$ in MetaAd.** To validate the effects of the hyper-parameter $\theta$ on the performance of MetaAd with $\varphi(x) = \frac{e^{\theta x} - 1}{e^{\theta} - 1}$, we give more details of the performances of MetaAd under different choices of $\theta$ in Fig. 4(a). We give both the empirical worst-case and average reward of MetaAd with different choices of $\theta$. The results show that the average reward of MetaAd is not significantly affected by the choice of $\theta$, but $\theta$ has a large effect on the empirical worst-case reward. This is because $\theta$ controls the conservativeness of MetaAd and hence is crucial for the worst-case competitive ratio when $\kappa \neq 0$ as discussed in Section 4.2. More specifically, a larger worst-case reward can be obtained with a smaller $\theta$ for the MovieLens dataset. The reason is that a higher level of conservativeness is needed when the maximum bid-budget ratio $\kappa$ is not zero.

**The effects of $\theta$ and $\lambda$ in LOBM.** The empirical worst-case and average rewards of LOBM with different choices of $\theta$ and $\lambda$ are provided in Fig. 4(b) and Fig. 4(c), respectively. Different choices of $\theta$ yield different competitive solution spaces, while the choices of $\lambda$ specify the relaxed robustness requirements of the worst-case competitive ratio for LOBM. Thus, we can get a different competitive ratio for LOBM by setting different $\theta$ and $\lambda$ as shown in Theorem C.1. These theoretical findings are validated by our numerical results.

As we can see from Fig. 4(b) and Fig 4(c), when $\lambda = 0$, the inequalities in the robust region (28) always hold, and hence LOBM reduces to pure ML and gives the same competitive ratio and average reward as ML. When $\lambda = 1$, LOBM guarantees the same competitive ratio as MetaAd, but does not necessarily always follow the solutions of MetaAd for each problem instance, since there exist other solutions that also satisfy the robustness requirement for certain problem instances. Therefore, when $\lambda = 1$, the competitive ratio and average reward of LOBM are close to but can be higher than those of MetaAd when the ML model used by LOBM is well trained. When $\lambda$ lies between 0 and 1, we can find that for some choices of $\theta$, LOBM can achieve an even better average reward than ML. This improvement comes from the fact that the competitive

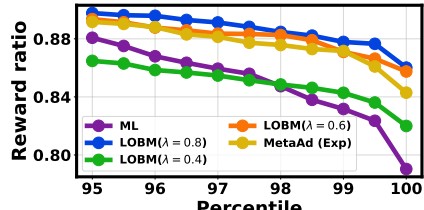

Figure 5: Reward (normalized by the offline optimal reward) at high percentiles (95% - 100%). $\theta$ is chosen as 1.

solution space in (28) can correct some low-quality ML predictions on certain problem instances. Also, for certain choices of $\theta$, LOBM can empirically achieve a better worst-case reward than MetaAd,

because `LOBM` can perform well due to ML predictions on some problem instances where `MetaAd` does not perform well. This observation validates that `LOBM` can effectively utilize the ML predictions to improve the average performance while guaranteeing a worst-case competitive ratio.

**Tail reward performance.** Last but not least, to evaluate the performance on adversarial/out-of-distribution instances, we show in Fig. 5 the reward (normalized by the offline optimal reward) at high percentiles from $95\%$ to $100\%$. We observe that the reward of `ML` quickly decreases when the percentile becomes higher and becomes the lowest at the high percentiles (larger than $98\%$), showing that `ML` is vulnerable to adversarial instances. Due to the worst-case competitiveness guarantees, `MetaAd` achieves a relatively higher reward even at high percentiles. Moreover, since `LOBM` guarantees the worst-case performance by the competitive solution space, the rewards of `LOBM` with different $\lambda$ are all higher than ML at high percentiles. The high percentile reward of `LOBM` increases with $\lambda$ because a larger choice of $\lambda$ guarantees a higher competitive ratio according to Theorem C.1. Interestingly, we can find that the rewards of `LOBM` at high percentiles are even larger than `MetaAd` when $\lambda$ is 0.6 or 0.8. This validates that when the ML model is well trained to provide high-quality predictions, `LOBM` can become more powerful and explore better matching decisions than the purely manual design of `MetaAd`.

### D.2 Online VM Placement

#### D.2.1 Problem setting

Virtual Machine (VM) placement is the process of matching the newly-created VMs to the most suitable servers in cloud data centers [13, 22, 26]. In this problem, once an end user send a VM request, the cloud operator needs to select a physical server for it. Different VM requests require different amount of physical computing resources. For example, the compute-optimized instances of Amazon EC2 [2] have different sizes, each requires different amount of computing resources. Due to the hardware heterogeneity [27], the available computing resources on different servers are different and the utilities of different servers can also be different. Our goal is to optimize the total utility of VM placement.

We consider a setup where the cloud manager allocates $V$ VMs (online nodes) to $U$ different physical servers (offline nodes). Based on the requirement of VMs, a VM request can be matched to a subset of the physical servers. The connections between VM requests and physical servers are represented by a bipartite graph $G$. A VM request $t$ at round $t, t \leq V$ has a computing load in the number of computing units denoted as $z_t$. Each server $u \in \mathcal{U}$ has a limited capacity of the computing units (e.g., virtual cores) denoted as $B'_u$. If the VM request $v$ is placed on a server $u$, the manager receives a utility proportional to the computing load $w_{u,t} = r_u \cdot z_t$ where $r_u$ is the utility of one computing unit on server $u$. Denoting $x_{u,t} \in \{0, 1\}$ as the decision on whether to place request $t$ on server $u$, the objective of the VM placement problem can be formulated as an OBM:

$$
\begin{aligned}
\max P &:= \sum_{t=1}^{V} \sum_{u \in \mathcal{U}} w_{u,t} x_{u,t} \\
\text{s.t. } &\forall u \in \mathcal{U}, \sum_{t=1}^{V} w_{u,t} x_{u,t} \leq B_u, \forall t \in [V], \sum_{u \in \mathcal{U}} x_{u,t} \leq 1, \forall u \in \mathcal{U}, v \in [V], x_{u,t} \geq 0,
\end{aligned}
\tag{41}
$$

where $w_{u,t} = r_u \cdot z_t$ and $B_u = r_u \cdot B'_u$. In this OBM, the VM request is not divisible, which means fractional matching is not allowed at any time and FLM does not apply.

#### D.2.2 Experiment setting

In the experiment, the cloud manager allocates $V = 100$ VMs (online nodes) to $U = 10$ different physical servers (offline nodes). We randomly generate graphs by Barabási–Albert method [3]. For an online node $v$, we sample its degree (the number of offline nodes connected to it) by a Binomial distribution $\mathcal{B}(U, d_v/U)$ where $d_v$ is the average degree of node $v$. The average degrees of online nodes are chosen from 4, 2, and 0.5. Each server has a capacity on the number of the computing units. The capacity $B'_u$ is sampled from a uniform distribution on the range $[20, 40]$. The computing load of a VM request is sampled from a uniform distribution on the range $[1, 4]$. The utility per computing unit $r_u$ is the price of a computing unit on the server $u$. We choose the price (in dollars) in the range $[0.08, 0.12]$ according to the prices of the compute-optimized instances on Amazon EC2

|  | Algorithms w/o ML Predictions | | | ML-based Algorithms | | | |
|---|---|---|---|---|---|---|---|
|  | Greedy | PrimalDual | MetaAd | ML | LOBM-0.8 | LOBM-0.5 | LOBM-0.3 |
| **Worst-case** | 0.6528 | 0.7937 | **0.8027** | 0.8005 | **0.8432** | 0.8253 | 0.8277 |
| **Average** | 0.6950 | 0.8343 | **0.8449** | **0.9626** | 0.934 | 0.9619 | 0.9610 |

Table 3: Worst-case and average rewards of different algorithms for VM placement. The worst-case and average rewards are normalized by optimal rewards. We compare `MetaAd` with the algorithms without using ML (Greedy and PrimalDual introduced in Section D.1.1). Additionally, we compare our learning-augmented algorithm `LOBM` with the ML algorithm. `LOBM`-$\lambda$ means `LOBM` with a slackness parameter $\lambda$ in Eqn. (28).

[2]. We randomly generate 20k, 1k, and 1k samples of BA graphs for training, validation and testing, respectively.

We compare our algorithms with baselines for OBM listed below. We compare our algorithms with the most common baselines for OBM as listed below.

- `OPT`: The offline optimal solution is obtained using Gurobi [11] for each graph instance.
- `Greedy`: The greedy algorithm [23] matches an online node to the available offline node that is connected to the node and has the highest bid value. `Greedy` has a strong empirical performance and is a special case of `MetaAd` with $\theta \to \infty$.
- `PrimalDual`: `PrimalDual` [23] calculates the scores of each bidder for each online node based on both the bid values and the remaining budgets, and then selects for an online node the available bidder with the highest score. It is a special case of `MetaAd` with $\theta \to 1$.
- `ML`: A policy-gradient algorithm that solves the OBM problem [1]. The inputs to the policy model are the available history information including the current bid value, the remaining budget of each offline node and the average matched bid value.

For learning-based algorithms, we use the neural networks which have two layers, each with 200 hidden neurons for fair comparison. The neural networks are trained by Adam optimizer with a learning rate of $10^{-3}$ for 50 epochs. Likewise, we use `LOBM`-$\lambda$ to refer to `LOBM` with a hyper-parameter $\lambda$ governing the competitiveness requirement in (28).

### D.2.3 Results

We first show Worst-case and average rewards of different algorithms for VM placement in Table 3. We observe that `MetaAd` achieves a higher worst-case and average reward than the the other algorithms without using ML (i.e., `Greedy` and `PrimalDual`). This is because `MetaAd` is more flexible to adjust the discounting function. Additionally, we can find that ML predictions can significantly improve the average performance compared to algorithms without ML predictions. In particular, `ML` even has an empirically higher worst-case reward than `Greedy` and `PrimalDual`, although it does not have a theoretical guarantee in terms of the worst-case competitive ratio. Note also that the worst-case reward ratio on the finite testing dataset is an empirical evaluation, and the true competitive ratio of `ML` without the theoretical guarantee can be even much lower than presented in the table. Importantly, we can find that `LOBM` can achieve a high average performance while guaranteeing a worst-case competitive ratio theoretically as shown in Theorem 5.1. Moreover, `LOBM` achieves the highest empirical worst-case reward among all the algorithms, because `LOBM` can effectively correct low-quality ML predictions for some difficult testing examples by learning augmented design and meanwhile also leverage good ML predictions to improve the performance for other testing examples.

**Effects of $\theta$ and $\lambda$.** Next, we give the ablation study of `MetaAd` and `LOBM` for different choices of parameters $\theta$ and $\lambda$ in Fig. 6. The parameter $\theta$ is the constant in the exponential discounting function and the parameter $\lambda$ is the slackness parameter in the competitive space in Eqn. (28). First, we give the worst-case and average rewards of `MetaAd` under different choices of $\theta$ in Fig. 6(a). We can find that compared with `PrimalDual` (i.e., $\theta = 1$), `MetaAd` can further improve the worst-case performance for the general bid settings by decreasing $\theta$ which is consistent with the competitive analysis in Corollary4.2.2.

Moreover, we provide the worst-case and average rewards for `LOBM` under different $\theta$ and $\lambda$ in Fig. 6(b) and Fig. 6(c), respectively. The results show that when $\lambda = 0$, `LOBM` reduces to pure `ML` and achieves the same worst-case and average performances as `ML`. The worst-case performance can

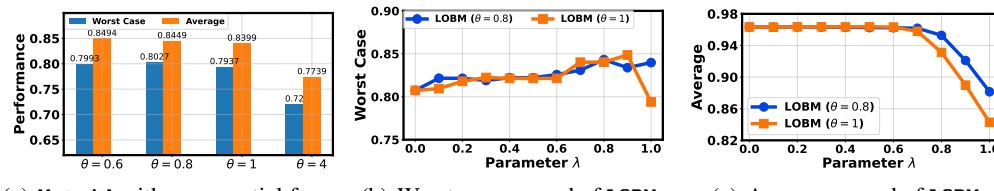

(a) `MetaAd` with exponential func-    (b) Worst-case reward of `LOBM`    (c) Average reward of `LOBM`
tion class

Figure 6: (a) Worst-case and average rewards of MetaAd with the exponential function class (a) Worst-case reward of LOBM. (b)Average reward of LOBM. The worst-case and average rewards are normalized by optimal rewards and are calculated empirically based on a testing dataset with 1000 samples.

be improved by increasing $\lambda$ since `LOBM` with a larger $\lambda$ has a higher competitive ratio guarantee according to Theorem C.1. However, the average performance can be affected when $\lambda$ becomes larger, because a larger $\lambda$ results in a smaller solution space for increased robustness and hence may exclude some solutions with high rewards for average cases. When $\lambda = 1$, `LOBM` shares the same theoretical competitive ratio bound as `MetaAd`, but it can still achieve better empirical worst-case and average rewards than `MetaAd` when the ML model is well trained.

