# OpenReview forum: "Online Budgeted Matching with General Bids"
_NeurIPS.cc/2024/Conference — NeurIPS 2024 poster_

### Official Review · Reviewer_3oQ3 · 2024-06-27

**Soundness:** 4
**Presentation:** 3
**Contribution:** 3
**Rating:** 7
**Confidence:** 5

**Summary:**

This paper addresses the Online Budgeted Matching (OBM) problem with general bids, which is fundamental in applications like online advertising and resource allocation. Traditional algorithms typically assume a very small bid-to-budget ratio κ, limiting their practicality. The authors remove the Fractional Last Matching (FLM) assumption, which allows for partial bids, and present a novel meta-algorithm called MetaAd. MetaAd is designed to adapt to different bid-to-budget ratios, achieving provable competitive ratios. The paper establishes an upper bound on the competitive ratio for any deterministic online algorithm. By introducing a general discounting function that adjusts based on the remaining budget, MetaAd matches offline nodes with the largest discounted scores. The algorithm’s effectiveness is demonstrated through theoretical proofs and numerical experiments, showing non-zero competitive ratios for κ in the range [0, 1). Additionally, the paper extends MetaAd to the FLM setting, achieving provable competitive ratios and improving the flexibility of the algorithm. Finally, the authors apply their competitive analysis to design a learning-augmented algorithm, LOBM, which leverages machine learning predictions to enhance average performance while maintaining a competitive ratio guarantee. The empirical validation on an online movie matching scenario demonstrates the practical benefits of the proposed approaches.

**Strengths:**

Originality
The paper stands out for its originality by addressing the Online Budgeted Matching (OBM) problem without relying on traditional assumptions like the small bid-to-budget ratio and the Fractional Last Matching (FLM) assumption. The introduction of the MetaAd algorithm is novel, offering a framework that adjusts to various bid-to-budget ratios while achieving provable competitive ratios. The extension of MetaAd to the FLM setting and the development of the learning-augmented algorithm, LOBM, further underscore its innovative contributions. The authors have done a commendable job of citing related work, clearly situating their contributions within the broader research landscape.

Quality
The submission is technically robust, featuring rigorous theoretical analysis and comprehensive proofs that establish the competitive ratios of the proposed algorithms. The authors provide clear upper bounds and validate MetaAd through detailed numerical experiments. Additionally, the empirical validation of the learning-augmented algorithm in an online movie matching scenario is well-executed, offering practical evidence to support the theoretical claims. The methods employed are well-suited to the problem, and the paper presents a complete and polished piece of work. The authors have been thorough and honest in evaluating both the strengths and limitations of their research.

Clarity
The paper is well-written and clearly organized, with a logical flow of ideas that is easy to follow. The introduction sets the stage effectively, and the related work section provides a comprehensive overview of prior research. The descriptions of the MetaAd algorithm and its extensions are detailed and precise, though some complex concepts could benefit from more intuitive explanations or visual aids. Overall, the writing style is concise and effective, ensuring that the reader is well-informed.

Significance
The findings are significant, addressing a critical problem in online budgeted matching. The proposed algorithms have direct applications in online advertising, resource allocation, and revenue management, making the results highly valuable for researchers and practitioners. By providing flexible and competitive algorithms that remove limiting assumptions, the paper advances the state of the art with substantial practical and theoretical improvements. The insights and methodologies are likely to inspire further research and development in this field.

**Weaknesses:**

Originality
While the paper introduces innovative approaches, its novelty primarily stems from the combination and adaptation of existing techniques rather than entirely new methodologies. The removal of the FLM assumption and the introduction of the MetaAd algorithm, although significant, build on a foundation of well-established concepts in online matching and competitive analysis. To enhance originality, the authors could explicitly discuss how their contributions fundamentally differ from prior work beyond the removal of specific assumptions. For instance, more emphasis on the unique aspects of their discounting function ϕ(x) and its derivation could underscore the novel elements of their approach.

Quality
The theoretical analysis is robust, but the empirical validation lacks breadth. The experiments focus on a single application scenario (online movie matching), which limits the generalizability of the results. Expanding the experimental evaluation to include a broader range of real-world applications would strengthen the paper’s claims and demonstrate the versatility of the proposed algorithms. For instance, additional experiments in online advertising or resource allocation contexts would provide a more comprehensive evaluation of MetaAd's performance. Moreover, while the paper provides clear proofs, some parts could be made more accessible by including step-by-step explanations, especially in complex derivations such as the proof of Proposition 4.1.

Clarity
Although the paper is generally well-organized, certain sections, particularly those involving complex theoretical concepts, could benefit from additional explanatory figures or diagrams. For example, visual aids illustrating the MetaAd algorithm's decision-making process and its competitive ratio calculations would enhance understanding. Additionally, some technical terms and mathematical notations could be more thoroughly explained to ensure accessibility to a broader audience. Clarifying the notation and providing more intuitive explanations for key concepts such as the discounting function ϕ(x) and its impact on the competitive ratio would help readers better grasp the core ideas.

Significance
The paper’s significance is somewhat limited by the scope of its empirical validation. While the theoretical contributions are substantial, the practical impact could be more convincingly demonstrated through a wider array of applications. Focusing on a single scenario limits the immediate relevance to other domains, potentially reducing the paper's broader appeal. To maximize its significance, the paper could include more diverse examples and discuss potential real-world implications in greater detail. For instance, illustrating how the MetaAd algorithm can be adapted and applied to various domains such as dynamic pricing or network routing would highlight its broader applicability and impact.

**Questions:**

1. In Section 2, you mention several existing algorithms for OBM. Could you provide a specific example comparing MetaAd’s performance and approach to one of these algorithms under the same conditions?
2. In the derivation of the discounting function ϕ(x) on page 5, could you include a step-by-step derivation or an illustrative example to clarify how ϕ(x) is constructed and why it is effective in maintaining competitive ratios?
3. In Section 4.2, you extend MetaAd to the FLM setting. Can you explain, with a specific example, how the algorithm adjusts when dealing with fractional bids? A flowchart or diagram showing this process would be helpful.
4. Your experiments focus on an online movie matching scenario. Could you provide results from a secondary domain, such as an online advertising campaign, where the budget constraints vary significantly? Including statistical metrics like precision and recall in this context would strengthen the empirical validation.
5. On page 8, you discuss the impact of κ on the competitive ratio. Could you include a graph showing how the competitive ratio changes with varying values of κ?
6. In your conclusion, you briefly touch on potential applications of MetaAd. Could you expand this section to include a detailed case study showing MetaAd’s implementation in a dynamic pricing model or a network routing problem?
7. Can you conduct a sensitivity analysis to show how changes in key parameters, such as the initial budget or the bid distribution, affect the algorithm’s performance?
8. Complex processes, such as the decision-making flow in MetaAd, could be better understood with visual aids. Could you create a figure or diagram that illustrates the algorithm’s workflow, including how decisions are made at each step in Appendix?

**Limitations:**

The authors acknowledge several limitations in their work. The authors did not explicitly discuss the potential negative societal impacts of their work. The paper does not discuss the potential impact on market competition. Favoring well-funded advertisers could lead to monopolistic behaviors and harm smaller competitors. Strategies to maintain a balanced competitive environment should be considered, such as setting caps on budget allocations or implementing measures to support smaller advertisers.

---

> ### Author Rebuttal · Authors · 2024-08-05
>
> We thank the reviewer for the questions.
>
> **`How does the contribution fundamentally differ from prior work?`**
>
> Beyond the removal of the small-bid and FLM assumptions, we present a novel meta algorithm (MetaAd) for general discounting functions. MetaAd can reduce to different competitive algorithms by choosing different discounting functions based on the meta competitive analysis (**Theorems 4.2 and 4.3**).  This lays a foundation for designing more complex scoring strategies for OBM with general bids. Additionally, our proof that uses several novel techniques (e.g., bounding primal-dual ratios and dual updates) and our learning-augmented algorithm design based on the competitive analysis also add to the literature and can be of independent interest.
>
> **`Experiments for other applications.`**
>
> We further validate our algorithms based on cloud resource management. The setup and results are available in our general rebuttal PDF. The results show that MetaAd achieves higher total utility than existing algorithms (Greedy and Primal Dual [1]). Also, with ML predictions, LOBM in Section 5 achieves a high average utility as well as a much better worst-case utility than a pure ML method without a competitive ratio guarantee.
>
> **`Q1 MetaAd’s performance and approach vs. existing algorithms`**
>
> Primal Dual [1] (equivalent to MSVV [2]) is an existing algorithm that uses a fixed exponential function as the discounting function. Greedy [1] is an existing algorithm that selects the offline node with the largest bid. By contrast, MetaAd selects different discounting function parameters for different general bid-budget ratios to maximize the competitive ratios. Our empirical results also demonstrate the practical advantage of MetaAd over these algorithms in various applications.
>
> **`Q2 How to construct $\phi(x)$ and why is it efficient in maintaining a competitive ratio`**
>
> Given any discounting function $\phi(x)$, we establish a competitive ratio for MetaAd in Theorem 4.2.  Thus, $\phi(x)$ is constructed to maximize the derived competitive ratio in Theorem 4.2 for different bid-budget ratios $\kappa$. To make this process tractable, we consider $\phi(x)$ within concrete parameterized function classes (e.g., exponential function class and quadratic function class) and optimize the competitive ratios by adjusting the parameters of the function class. In this way, MetaAd achieves a high competitive ratio by using different parameterized discounting functions for different $\kappa$. We observe that MetaAd with exponential function class is appealing to maintain a high competitive ratio (best-known without FLM).
>
> **`Q3 How to deal with fractional bids?`**
>
> MetaAd with FLM is given in Algorithm 3. The key adjustment is the scoring strategy for the offline nodes with insufficient budgets. Without FLM, MetaAd scores the offline nodes with insufficient budget as zero to avoid selecting them. Differently, FLM sill allows to match a query to an offline node with an insufficient budget, and the matched offline node pays the remaining budget $b_{u,t-1}$ up to the bid value. Thus, we score the offline nodes with insufficient budgets according to the remaining budget $b_{u,t-1}$ instead of scoring them as zero. In this way, the scoring is based on the true payment in the FLM setting. This contributes to a higher competitive ratio than the non-FLM setting.
>
> **`Q4 Results from a secondary domain`**
>
> We provide another set of empirical results on cloud resource management in the rebuttal PDF and further validate the benefits of our algorithms. The budget constraints in this application are very different from those in online movie matching in terms of the range of the initial budgets and bid values.
>
> **`Q5 A figure to show the impact $\kappa$?`**
>
> On Pages 7 & 8, we have figures showing the competitive ratios varying with $\kappa$. (Figure 1 is for non-FLM and Figure 2 is for FLM).
>
> **`Q6 A detailed case study showing MetaAd’s implementation...`**
>
> We have provided a detailed case study on online movie matching in Section D.1. Additionally, we implement MetaAd on cloud resource management with results given in the rebuttal PDF. The case studies both validate the superior performance of MetaAd compared with existing deterministic algorithms [1,2]. We’ll also include other applications (e.g., network routing per the reviewer’s suggestion) in the final version.
>
> **`Q7 Sensitivity analysis for the impact of key parameters`**
>
> In cloud resource management in the rebuttal PDF, the range of initial budget and the bid distribution are different from those in movie matching in Section D.1.  MetaAd achieves the best performance among the competitive algorithms (Greedy and Primal Dua [1]). Additionally in the rebuttal PDF, we give the sensitivity study showing the performance of MetaAd varying with $\theta$ in the discounting function and the performance of LOBM varying with $\lambda$. The same sensitivity study is included in Fig. 3 in Section D.1.2.
>
> **`Q8 A diagram to illustrate the algorithm’s workflow?`**
>
> Thanks for the suggestion. We’ll include a diagram illustrating the algorithm's workflow in the appendix in our revision.
>
> **`Potential negative societal impacts`**
>
> Thanks for the suggestion. For online advertising, if there is a large disparity of the initial budgets among advertisers, those with a larger initial budget may be matched with a larger chance due to their smaller bid-budget ratios. This fairness issue exists in prior algorithms [1,2] and warrants further discussions. We’ll discuss this societal impact in the revision.
>
> **`Reference`**
>
> [1] Mehta, A., 2013. Online matching and ad allocation. Foundations and Trends® in Theoretical Computer Science, 8(4), pp.265-368.
>
> [2] Mehta, A., Saberi, A., Vazirani, U. and Vazirani, V., 2007. Adwords and generalized online matching. Journal of the ACM (JACM), 54(5), pp.22-es.

---

> > ### Author Response · Authors · 2024-08-13
> >
> > Dear Reviewer,
> >
> > Thank you very much for taking the valuable time to review our paper. We hope our responses have satisfactorily addressed your concerns. As the discussion deadline approaches, we are more than pleased to answer any remaining questions you may have.

---

> > > ### Comment · Reviewer_3oQ3 · 2024-08-13
> > >
> > > Thank you to the authors for their thorough rebuttal, which has addressed most of my concerns. I will maintain my positive rating and raise my confidence level to 5.

---

> > > > ### Author Response · Authors · 2024-08-13
> > > >
> > > > Thank you very much for reading our rebuttal. We're glad that we have addressed your concerns.

---

### Official Review · Reviewer_CTTb · 2024-06-30

**Soundness:** 3
**Presentation:** 2
**Contribution:** 2
**Rating:** 4
**Confidence:** 4

**Summary:**

This paper studies the online budgeted matching problem without the small-budget or Fractional Last Matching (FLM) assumption. The authors propose a meta algorithm called MetaAd, which uses a discount function to assign each node with a score. The authors perform competitive analysis for their meta algorithm and show how the algorithm performs under different discount functions. They also provide theoretical competitive ratios for the FLM setting, and a learning-augmented setting.

**Strengths:**

- The theoretical results of this paper appear sound.
- I appreciate that the authors take efforts in presenting their results in a unified framework, establishing connection with the competitive analysis for small-bid setting, as well as prior works such as BJN2007.
- The authors provided a good summary of prior works studying the same problem (under other assumptions).

**Weaknesses:**

1. My main concern for this work lies in its contribution. More specifically,
- I feel that the main contribution of the meta algorithm is in the introduction of the discounting function $\phi$ and using it to regulate the matching decision. However, as the authors suggest, how to choose the discounting function is unclear. Without a clear understanding of how to choose the best $\phi$, the proposed algorithm will be much less practical.
- Following the points above, I also find the competitive ratios provided in Theorem 4.2 and Corollary 4.2.2, 4.2.3 a bit difficult to digest. While the authors provided numerical evaluations of these competitive ratios, it is still unclear why the authors decided to pick the functional form $C(\exp(\theta x) -1)$ and $C x^2$ and how the constants $C$ are determined. As a result, evaluating how well the meta algorithm actually performs become less straightforward.
- Since the main proof techniques come from primal-dual analysis, I also wonder whether the proof also contains technical novelty.

2. No numerical experiments are provided in the main body of the paper. On a related note, I also have concerns over the practicality of such an algorithm to real-world ads systems due to (i) lack of knowledge of the discounting function; (ii) scalability of the problem, especially when the node set can have enormous size. I wonder if the authors might also comment on these.

**Questions:**

See weakness.

**Limitations:**

See weakness.

---

> ### Author Rebuttal · Authors · 2024-08-02
>
> We thank the reviewer for the questions.
>
> **`How to choose a discounting function $\phi$?`**
>
> Our results (**Theorem 4.2 and 4.3** ) are instrumental to design a discounting function $\phi$. Specifically, Theorem 4.2 and 4.3 enable us to identify appropriate $\phi$ by maximizing the derived competitive ratios. In this work, we focus on two important function classes -- exponential function class and quadratic function class --- and solve the best $\phi$ within the two function classes. We find that even the simple exponential function class can offer appealing results: It recovers the optimal competitive ratio of $1-1/e$ in the small-bid setting, gets close to the upper bound of the competitive ratio for large bids $\kappa$ without FLM (**Figure 1**), and matches or approaches the best-known competitive ratios of deterministic algorithms ( [BJN2007] for small enough bids and Greedy for large bids ) with FLM (**Figure 2**).
>
> Additionally, our results provide a foundational guide for more complex designs in OBM. Specifically, our learning-augmented design LOBM in **Section 5** is a successful example of applying our theoretical results to improve average performance while offering competitive ratio guarantees. In this example, we use an ML model to explore a discounting function and make sure it lies in a discounting function space defined by our competitive analysis. In the future, it’s interesting to search for discounting functions over more complex function classes, either analytically or by AI techniques based on our theoretical analysis.
>
> **`Why use the exponential function forms and how to determine the constants?`**
>
> As our analysis shows, the discounting function $\phi(x)$ must be a positive decreasing function of the remaining budget because:
> + Intuitively, the decreasing discount function reduces the chance of selecting the offline nodes with less remaining budget, due to the scarcity of the remaining budget.
> + Theoretically, a positive decreasing function  $\phi(x)$  is required to satisfy the dual feasibility (Lemma 1).
>
> We consider $\varphi(x)=1-\phi(1-x)$ with the exponential class $C(e^{\theta x}-1)$ or quadratic class $Cx^2$ as examples. Given a bid-to-budget ratio $\kappa$, the constants can be easily determined by maximizing the competitive ratios (Eqn. (5) for exponential class or Eqn. (6) for quadratic class).
>
> **`Technical novelty`**
>
> Due to our novel and generalized settings, our proof uses several novel techniques as summarized below.
>
> + **Upper bound the competitive ratio for general bids with no FLM (Proposition 4.1)**.  OBM with general bids with no FLM has many key applications (e.g. cloud resource management) but is not well studied theoretically. By a difficult example, we upper bound the competitive ratio for the first time and show the difficulty of this problem theoretically.
> + **Dual construction for general bids**. With small bids, the remaining budget is almost zero when an offline node has insufficient budget to match a query, but the remaining budget can be large when budget insufficiency happens for general bids.  This presents new challenges in guaranteeing the dual feasibility. We give a new dual construction (*Algorithm 2* for non-FLM, and *Algorithm 4* for FLM) where dual variables are set based on the remaining budget and adjusted at the end of the algorithm to satisfy the dual feasibility with small enough dual increment.
> + **Techniques to bound the primal-dual ratio for general bids with no FLM (Theorem 4.2)**. It is a key step to bound the primal-dual ratio which translates to the competitive ratio by Lemma 1. The challenges come from the unspecified discounting function $\phi$ and the absence of small-bid and FLM assumptions.   Without specific forms of $\phi$, we derive a sufficient condition of guaranteeing a primal-dual ratio and the condition gives us the primal-dual ratio bound. To get the condition, we bound the discrete sum of dual variables with general bids by an integral (Eqn.(15)). Besides, we bound the dual increment due to dual feasibility for non-FLM setting (Eqn. (19)).
> + **Techniques to bound the primal-dual ratio for general bids with FLM (Theorem 4.3)**. With the FLM assumption, the dual adjustment to satisfy the dual feasibility is different from the non-FLM setting. Thus, we extend our techniques to FLM by bounding the dual increment due to dual feasibility (Eqn. (26)).
> + **Robustness analysis for learning-augmented OBM (Theorem 5.1).**  Different from the existing learning-augmented OBM [1,2] that relies on an existing competitive baseline, we design a discounting function space (Eqn. (28)) based on our competitive analysis. This leads to LOBM which guarantees a competitive ratio given any ML model without a baseline as input.
>
> **`Practicality of such an algorithm in real-world systems`**
> + **Choice of discounting function**. As stated in responses above, the exponential function with optimal constants can give us very good competitive guarantees (best-known for general bids with no FLM). To further improve the average performance under the competitive guarantee, we can apply LOBM in Section 5. The empirical results in Section D.1.2 and rebuttal PDF show the superiority of our algorithms in both average and worst-case performance.
> + **Scalability**. For both MetaAd (Algorithm 1) and LOBM (Algorithm 5), whenever an online node arrives, the scores for offline nodes can be calculated efficiently in parallel by each offline node, and then the maximally scored offline node is matched subject to budget availability. Therefore, like the prior OBM algorithms, our algorithms can easily scale with the size of the offline node set.
>
> **`Reference`**
>
> [1] Choo, D., Gouleakis, T., Ling, C.K. and Bhattacharyya, A., 2024. Online bipartite matching with imperfect advice. In ICML.
>
> [2] Li, P., Yang, J. and Ren, S., 2023, July. Learning for edge-weighted online bipartite matching with robustness guarantees. In ICML.

---

> > ### Author Response · Authors · 2024-08-13
> >
> > Dear Reviewer,
> >
> > Thank you very much for taking the valuable time to review our paper. We hope our responses have satisfactorily addressed your concerns. As the discussion deadline is about to conclude, we are more than pleased to answer any remaining questions you may have.

---

### Official Review · Reviewer_vF5Z · 2024-07-09

**Soundness:** 4
**Presentation:** 3
**Contribution:** 3
**Rating:** 7
**Confidence:** 3

**Summary:**

The paper studies the classic online budgeted matching problem (OBM), relaxing the small bid assumption and the FLM assumption. Precisely, an upper bound on the competitive ratio is proven for any deterministic algorithms. Then a framework of algorithms for OBM is proposed to solve OBM with general bids, which uses various discounting function to represent the degree of budget insufficiency given a bid-budget ratio. Correpondingly, the competitive ratios of the algorithms are proved. Finally, the learning augmented algorithm, LOBM, is extended under the FLM assumption, achieving better performance..

**Strengths:**

1. The novel design of the meta algorithms for OBM, MetaAd, is interesting and inspiring.
2. Concrete theoretical proofs for all the results.
3. Well-organized presentation.

**Weaknesses:**

1. In the motivation scenario, such as online advertising, online service matching, revenue management, the small bid assumption usually holds. This means relaxing such an assumption may be not much motivative.

**Questions:**

NA

---

> ### Author Rebuttal · Authors · 2024-08-02
>
> Thank you for your question regarding the motivation of general `non-small` bids.
>
> Relaxing the small bid assumption is crucial to providing provable algorithms for many useful online bipartite matching (OBM) scenarios. We explain the significance from both application and theory sides.
>
> + **Applications to broader scenarios.** Although in some scenarios the bid-to-budget ratio is small enough to justify the previous small-bid assumption, many applications have "medium bids", where the bid-to-budget ratios are neither negligible nor as large as 1. Take cloud virtual machine (VM) allocation as an example: One VM request can commonly take up a non-negligible fraction (e.g., 1/10 to 1/2) of the total computing units in a server, which is equivalent to a non-small bid setting.
>
> + **Bridge the gap in theory.** The small-bid assumption is restrictive in the sense that it assumes the bid-budget-ratio is infinitesimally small ($\kappa \rightarrow 0$) and only models a limited set of online bipartite matching problems.  By relaxing the small bid assumption, OBM with general bids covers a broader set of online matching problems, including the vertex-weighted online bipartite matching [1]. Thus, compared to the small-bid setting, our study provides provable algorithms for more generalized online bipartite matching problems.
>
> In fact, some recent works have highlighted the importance of generalizing OBM from small-bid to general-bid settings [1,2]. Some of them provide provable algorithms for settings with the fractional last matching (FLM) assumption [2]. Importantly, we design meta algorithms for settings both with and without the FLM assumption.
>
>
> **`Reference`**
>
> [1] Mehta, A., 2013. Online matching and ad allocation. Foundations and Trends® in Theoretical Computer Science, 8(4), pp.265-368.
>
> [2] Huang, Z., Zhang, Q. and Zhang, Y., 2020, November. Adwords in a panorama. In 2020 IEEE 61st Annual Symposium on Foundations of Computer Science (FOCS) (pp. 1416-1426). IEEE.

---

> > ### Author Response · Authors · 2024-08-13
> >
> > Dear Reviewer,
> >
> > Thank you very much for taking the valuable time to review our paper. We hope our responses have satisfactorily addressed your concerns. As the discussion deadline approaches, we are more than pleased to answer any remaining questions you may have.

---

> > ### Comment · Reviewer_vF5Z · 2024-08-13
> >
> > Thanks for your response.

---

### Official Review · Reviewer_JnBx · 2024-07-11

**Soundness:** 4
**Presentation:** 3
**Contribution:** 1
**Rating:** 6
**Confidence:** 3

**Summary:**

**Problem Studied**

This paper studies the online budgeted matching problem (also known as AdWords). The input to the problem is a bipartite graph, where one side of the graph (the advertisers) is known in advance. Each advertiser has a budget, which is the maximum amount of money they are able to spend. The nodes on the other side of the graph (the ad impressions) arrive one by one in an online manner. When an online node arrives, each advertiser submits a bid, which is the amount they are willing to pay for the ad impression. The algorithm then irrevocably decides which advertiser to match the impression to, and the advertiser pays their bid. The goal of the algorithm is to maximize the total amount of money that is spent.

**Main Results/Contributions**

Adwords has been most commonly studied under a "small-bids" assumption, which essentially states that the ratio of the bid of any advertiser to their budget is small. Under this assumption, there are deterministic algorithms which achieve a $1-\frac{1}{e}$ competitive ratio. This paper studies Adwords under general bids. The main results are:
1. An upper bound on the competitive ratio of any deterministic algorithm parameterized by the bid-budget ratio, and
2. A "meta algorithm" which is parameterized by a discounting function $\phi$, and a competitive ratio bound for the algorithm that is parameterized by some properties of $\phi$ and the bid-budget ratio.

**Strengths:**

This paper is generally well-written and easy to understand. I believe the main results of the paper to be correct, although the only proof I checked in detail is the proof of the upper bound for deterministic algorithms.

**Weaknesses:**

The main criticism I have of the paper is that it is unclear why it is useful to study Adwords in the setting with general bids and without the FLM assumption. The setting of general bids makes sense to me. However (and please correct me if I am wrong), it seems like the FLM assumption is the same as assuming that the bid of any advertiser cannot exceed their remaining budget. This assumption seems very reasonable to me; why would any advertiser submit a bid that is greater than their remaining budget? In any case, since the algorithm knows the total budgets of the advertisers, it could just prevent an advertiser from bidding above their remaining budget.

**Questions:**

1. Why is it interesting to study the setting with general bids and no FLM? (see above)
2. In the paper "AdWords in a Panorama" [10], the authors give a 0.5016-competitive algorithm for Adwords with general bids. However, since the current paper claims to be the first to give a competitive algorithm for Adwords with general bids and without the FLM assumption, it must be the case that [10] uses the FLM assumption. Can you clarify where [10] needs to use the FLM assumption? I think this would be useful to understand the importance of the FLM assumption.

**Limitations:**

Yes.

---

> ### Author Rebuttal · Authors · 2024-08-01
>
> Thank you for your questions.  We answer them as below.
>
> **`Why is it interesting to study the setting with general bids and no FLM?`**
>
> OBM with general bids covers a wide range of online bipartite matching problems [1], so it models many applications where FLM does not hold. Two examples are given below.
>
> + **Cloud resource management [2,3].**  In this problem, the cloud manager needs to allocate virtual machines (VMs, online nodes) to different physical servers (offline nodes).  Each server $u$ has a computing resource capacity of $B_u'$. Whenever a VM request $v$ arrives, the manager observes its computing load, denoted as $z_{v}$. If the VM request is placed on a server $u$, the manager receives a utility proportional to the computing load $w_{u,v}= r_u\cdot z_{v}$ given the heterogeneity of servers.  Denoting $x_{u,v}\in$ { 0,1}  as whether to place $v$ to VM $u$, the goal is to maximize the total utility $\sum_{v=1}^V\sum_{u=1}^U w_{u,v}x_{u,v}$ subject to the computing resource constraint for each server $u$:  $\sum_{v=1}^V z_v x_{u,v}\leq B_u'$ which can also be written as $\sum_{v=1}^V w_{u,v} x_{u,v}\leq B_u$ with $B_u=B_u'\cdot r_u$. In this OBM problem, the VM request is not divisible, i.e., fractional matching is not allowed at any time.
>
> + **Inventory management with indivisible goods.** In this problem, a manager needs to match several indivisible goods (which arrive at different times) to different resource nodes each with limited capacity (e.g., matching parcels to mail trucks, matching food orders to delivery vehicles/bikes). Each good can only be matched to one node without splitting. A good $v$ can take up a percentage of $w_{u,v}$ for resource node $u$. The target is to maximize the total utilization $\sum_{v=1}^V\sum_{u=1}^U w_{u,v}x_{u,v}$ subject to the capacity constraint at each node $\sum_{v=1}^V w_{u,v} x_{u,v}\leq 1$.
>
> By studying OBM with general bids and no FLM, we provide provable algorithms for broader applications beyond online advertising (whether FLM holds in online advertising depends on the specific policies of the advertising platforms).
>
> Additionally, we also extend MetaAd to OBM with general bids and FLM (see **`Appendix B`**). The algorithm and analysis are adjusted to exploit the benefits of FLM assumption. By assigning the discounting function with an exponential function class, MetaAd gives high enough competitive ratios for any bid-to-budget ratios compared to existing deterministic algorithms. Thus, for the FLM setting, our MetaAd is still effective in building competitive algorithms for general bids.
>
> **`Clarify where [10] needs the assumption of FLM`**
>
> In the first paragraph of the Introduction section in [10], the authors claim the use of the FLM assumption: The platform selects an advertiser for an online impression and pays either its bid or its remaining budget, whichever is smaller.
>
> FLM assumption is necessary in [10]. First, the algorithms and analysis in [10] rely on the reformulation of OBM as a Configuration Linear Program. The objective of Configuration LP depends on the budget-additive payment which means each advertiser pays the sum of the bids for all the online nodes assigned to it or up to its total budget, whichever is smaller. Thus, when the sum of the assigned bids is larger than the available budget for an advertiser, the matching is fractional.
> Besides, whenever an online node arrives,  the basic algorithm (Algorithm 1) of [10] assigns an offline node to an online node no matter whether the offline node has a sufficient remaining budget or not. In this algorithm, the FLM assumption is necessary to make sure the total payment of each advertiser does not exceed the initial budget.
>
> Moreover, [10] provides a randomized algorithm, which assigns offline nodes to online nodes with randomness. By contrast, we focus on deterministic algorithms and make novel contributions to OBM by considering general bids for settings both with and without FLM.
>
> **`Reference`**
>
> [1] Mehta, A., 2013. Online matching and ad allocation. Foundations and Trends® in Theoretical Computer Science, 8(4), pp.265-368.
>
> [2] Grandl, R., Ananthanarayanan, G., Kandula, S., Rao, S. and Akella, A., 2014. Multi-resource packing for cluster schedulers. ACM SIGCOMM Computer Communication Review, 44(4), pp.455-466.
>
> [3] Speitkamp, B. and Bichler, M., 2010. A mathematical programming approach for server consolidation problems in virtualized data centers. IEEE Transactions on services computing, 3(4), pp.266-278.

---

> > ### Comment · Reviewer_JnBx · 2024-08-12
> >
> > Thank you for the detailed response! It is very helpful and I appreciate it. The authors have addressed my main question, and I have decided to increase my score to a 6.

---

> > > ### Author Response · Authors · 2024-08-12
> > >
> > > Thank you very much for taking the time to consider our responses! We’re glad to have addressed your concerns, and we appreciate your decision to increase the score. Please don’t hesitate to let us know if you have any remaining questions.

---

### Author Rebuttal · Authors · 2024-08-05

We thank all the reviewers for their valuable comments and questions. We've included results for a new set of experiments based on cloud resource management. The results are available in the attached PDF.

**`Experiment setup`**

In the experiment, the cloud manager allocates 100 virtual machines (VMs, online nodes) to 10 different physical servers (offline nodes).  A VM request can be matched to a subset of the physical servers. The connections between VM requests and physical servers are represented by a bipartite graph. We randomly generate graphs by Barabási–Albert method [1] with the average degree of online nodes chosen from 4, 2, and 0.5.  We consider a setting where the limited resource is the computing units (e.g., virtual cores) [2]. Each server $u$ has a capacity of $B_u'$ computing units and $B_u'$ is sampled from a normal distribution with a mean of 40.  The computing load of a VM request $v$ is the number of the requested computing units, denoted as $z_{v}$, which is sampled from a uniform distribution on the range 1 to 8.  If the VM request is placed on a server $u$, the manager receives a utility proportional to the computing load, $w_{u,v}= r_u\cdot z_{v}$ with $r_u$ being the price of each computing unit of server $u$. We choose the price $r_u$ (in dollars) in the range [0.08,0.12] according to the prices of the compute-optimized instances on Amazon EC2 (a large cloud service provider).

Denoting $x_{u,v}\in $ { 0,1}  as whether to place $v$ to VM $u$, the goal is to maximize the total utility $\sum_{v=1}^V\sum_{u=1}^U w_{u,v}x_{u,v}$ subject to the computing resource constraint for each server $u$:  $\sum_{v=1}^V z_v x_{u,v}\leq B_u'$ which can also be written as $\sum_{v=1}^V w_{u,v} x_{u,v}\leq B_u$ with $B_u=B_u'\cdot r_u$.  We randomly generate 10k, 1k and 1k graph samples for training, validation and testing, respectively. We use a neural network as the ML model in ML-based algorithms. The neural networks in different ML algorithms have two layers, each with 200 hidden neurons for fair comparison. The neural networks are trained by Adam optimizer with a learning rate of 0.001 for 50 epochs.

**`Reference`**

[1] Borodin, A., Karavasilis, C. and Pankratov, D., 2020. An experimental study of algorithms for online bipartite matching. Journal of Experimental Algorithmics (JEA), 25, pp.1-37.

[2] Grandl, R., Ananthanarayanan, G., Kandula, S., Rao, S. and Akella, A., 2014. Multi-resource packing for cluster schedulers. ACM SIGCOMM Computer Communication Review, 44(4), pp.455-466.

---

### Decision · Program_Chairs · 2024-09-25

**Decision:**

Accept (poster)

**Comment:**

Majority of reviewers are positive about the paper. They think the paper's idea is novel, theoretical results are sound, and the paper is well-written. The reviewers do have some concerns on the practical applicability of the results, and I have similar concerns. The paper could be further improved if these concerns can be addressed.